# Reasoning on Time-Series for Financial Technical Analysis

**Kelvin J.L. Koa[1][\*][†] Jan Chen[2][\*] Yunshan Ma[3] Huanhuan Zheng[4] Tat-Seng Chua[1]**
[1]National University of Singapore  [2]Technical University of Munich
[3]Singapore Management University  [4]City University of Hong Kong
`kelvin.koa@u.nus.edu, jan.c.chen@tum.de, ysma@smu.edu.sg,`
`h.zheng@cityu.edu.hk, dcscts@nus.edu.sg`

## Abstract

While Large Language Models have been used to produce interpretable stock forecasts, they mainly focus on analyzing textual reports but not historical price data, also known as Technical Analysis. This task is challenging as it switches between domains: the stock price inputs and outputs lie in the time-series domain, while the reasoning step should be in natural language. In this work, we introduce Verbal Technical Analysis (VTA), a novel framework that combine verbal and latent reasoning to produce stock time-series forecasts that are both accurate and interpretable. To reason over time-series, we convert stock price data into textual annotations and optimize the reasoning trace using an inverse Mean Squared Error (MSE) reward objective. To produce time-series outputs from textual reasoning, we condition the outputs of a time-series backbone model on the reasoning-based attributes. Experiments on stock datasets across U.S., Chinese, and European markets show that VTA achieves state-of-the-art forecasting accuracy, while the reasoning traces also perform well on evaluation metrics judged by industry experts. Our code is available at: `https://github.com/chen-jan/VTA`.

## 1 Introduction

With the advent of Large Language Models (LLMs), an increasingly popular application is in financial analysis (Wu et al., 2023; Xie et al., 2023). This spans a wide range of tasks, including financial question answering (FinQA) (Liu et al., 2025b; Qian et al., 2025), investment decision-making (Yu et al., 2025; 2024), and market forecasting (Yu et al., 2023; Koa et al., 2024). Majority of existing approaches primarily utilize the strong natural-language capabilities of LLMs to analyze financial reports or do sentiment analysis on social texts (see Table 1), but neglects interpretable analysis on stock price data, which arguably contain useful information for financial practitioners.

The current solutions from the general time-series domain are not yet sufficient for this task. Existing studies on time-series reasoning (Merrill et al., 2024; Chow et al., 2024) consistently report that LLMs struggle to reason over raw time-series inputs. Meanwhile, time-series LLMs (Jin et al., 2024; Liu et al., 2025a) often rely on reprogramming the embedding space, which produces time-series outputs but sacrifices verbal reasoning ability, which is an essential requirement for interpretable financial analysis. The closest effort is TimeCAP (Lee et al., 2025), which generates explanations by contextualizing the series with auxiliary information. However, its reasoning trace is derived from *external* data, and it produces classification label forecasts rather than full time-series trajectories.

Unlike other time-series data, financial time-series contains *intrinsic* interpretable signals which are widely studied by experts, known as Technical Analysis (Kirkpatrick II & Dahlquist, 2010). We use these signals to verbally analyze financial time-series and produce interpretable stock forecasts.

The use of LLMs for time-series reasoning is hindered by three main challenges. Firstly, current LLMs have limited capabilities in time-series forecasts. Some works have tackled this by modifying the embedding space to produce time-series outputs (Jin et al., 2024; Liu et al., 2025a), but this comes at the cost of interpretability as the LLM loses its natural language capability. Secondly, on a

---

* Equal contribution.  † Corresponding author.

Table 1: Comparison of relevant works. Our work contributes a novel explainable financial signal for practitioners and produces some insights into how time-series forecasting can be made interpretable.

| Models | Domain | Input | Output |
|---|---|---|---|
| **Financial LLMs** | | | |
| Fin-R1 (Liu et al., 2025b), Fino1 (Qian et al., 2025) | Financial Question Answering | Financial Reports | Textual Answers |
| FinMem (Yu et al., 2025), FinCon (Yu et al., 2024) | Investment Decision-Making | Text, Tabular, Audio | Binary (Buy/Sell) |
| GPT-4 (Yu et al., 2023), SEP (Koa et al., 2024) | Market Direction Forecasting | News, Social Texts | Binary or Quantile-Based |
| **Time-Series LLMs** | | | |
| Time-LLM (Jin et al., 2024), CALF (Liu et al., 2025a) | Time-Series Forecasting | General Time-Series | Time-Series |
| TimeCAP (Lee et al., 2025) | Time-Series Reasoning | Time-Series + Auxiliary | Classification + Reasoning |
| **Ours** | | | |
| Verbal Technical Analysis (VTA) | Time-Series Reasoning (Financial) | Financial Time-Series | Time-Series + Reasoning |

higher level, current LLMs are not known to have the ability to do verbal reasoning on time-series to produce accurate forecasts (Merrill et al., 2024; Chow et al., 2024). This involves understanding how to best analyze the predictive signals in the time-series data in an unsupervised manner. Thirdly, the reasoning trace of the LLM further needs to be converted into time-series output to produce useful stock forecasts. LLMs are typically fine-tuned on next-token predictions (Radford et al., 2019), and using direct time-series outputs would not produce the best forecasts, which we verify empirically.

To address these problems, we present three key contributions. Firstly, we propose our Verbal Technical Analysis (VTA) framework, which combines a backbone time-series model (which we termed as "latent thinking") with a reasoning LLM (termed as "verbal reasoning") to produce interpretable stock time-series forecasts. This framework combines the strong pattern processing ability of state-of-the-art time-series models and the strong reasoning ability of LLMs to produce forecasts that are both accurate and interpretable. Secondly, for reasoning over time-series, the stock time-series data is converted into textual annotations (Lin et al., 2024) as inputs to the LLM. The reasoning trace is then optimized through a modified Group Relative Policy Optimization (GRPO) objective (Shao et al., 2024) that uses an inverse Mean Squared Error (MSE) reward scoring, which we termed as Time-GRPO. Thirdly, to produce time-series outputs from the reasoning traces, we condition (Ho & Salimans, 2022) the generated outputs from the time-series model on the reasoning-based attributes.

To demonstrate the effectiveness of VTA, we perform extensive experiments across established stock baselines (Xu & Cohen, 2018) and additional stock data across the U.S., Chinese, and European markets. We show that our model forecasts achieve state-of-the-art results in prediction accuracy, while also being interpretable. In addition, evaluation by industry experts show that the reasoning traces score high across various evaluation metrics from literature. Finally, to justify the practical capability of the model, we form Markowitz portfolios across the prediction length and show that the portfolios formed from VTA forecasts can also perform well on investment metrics.

## 2 RELATED WORKS

**Financial Large Language Models.** The rise of Large Language Models (LLMs) has spurred a growing body of research on their application in finance. The earliest works focus on developing general-purpose financial LLMs, such as BloombergGPT (Wu et al., 2023) and FinMA (Xie et al., 2023), by finetuning on a large set of financial corpora across multiple downstream tasks. Later works began to tackle the specific challenges of LLMs in finance. For example, works on financial question-answering (FinQA) (Liu et al., 2025b; Qian et al., 2025) focus on teaching LLMs to analyze financial reports, which requires the ability to read structured financial tables and extract insights from complex documents (Zhu et al., 2021). LLMs for investment decision-making (Yu et al., 2025; 2024) typically utilize multi-agent systems to handle different parts of an investment decision, including but not limited to document analysis, memory, risk control, *etc.* LLMs for financial forecasting (Yu et al., 2023; Koa et al., 2024) seeks to predict the direction in which the market will go. Typically, these works analyze textual sources in order to understand the sentiment or financial health of a company. Our work positions itself in this field by learning to produce interpretable forward-looking signals from financial time-series data, which could benefit all applications above.

**Time-Series Large Language Models.** At the same time, there is also a growing body of research in utilizing LLMs in the time-series domain (Kong et al., 2025). LLMs for time-series (Jin et al., 2024; Liu et al., 2025a) leverage the large scale parameters and robust pattern recognition of LLMs

by fine-tuning them for time-series forecasting tasks. However, these approaches typically modify the embedding space of the LLMs, making them lose their original language reasoning capabilities. Some works have also explored the ability of LLMs to reason over time-series data. It was found that language models are "remarkably bad" at zero-shot time-series reasoning (Merrill et al., 2024), whereas fine-tuning them using a latent encoder (Chow et al., 2024) shows some promising early results in reasoning over time-series for captioning. The closest work on time-series reasoning comes from TimeCAP (Lee et al., 2025), which perform forecasting by contextualizing the input time-series with auxiliary time-series information. It produces explanations by searching for similar historical contexts, and produces classification label forecasts, in textual form. Our work builds on this line of research by exploiting the predictive signals within financial time-series to produce interpretable time-series forecasts, providing some insights on the capabilities of LLMs in reasoning over time-series data.

## 3 VERBAL TECHNICAL ANALYSIS

The Verbal Technical Analysis (VTA) framework is shown in Figure 1. There are three components: **(1)** In Time-Series Reasoning, we teach an LLM to verbally reason over the time-series inputs. This is done through a textual annotator to extract useful indicators, and a proposed Time-Series Group Relative Policy Optimization (Time-GRPO) method; **(2)** In Time-Series Forecasting, we train a backbone forecasting model, which can better learn from the complex low-level patterns in the time-series data; **(3)** In Joint Conditional Training, the time-series forecast is conditioned on the reasoning attributes, and the model is trained over the conditional and unconditional forecasts concurrently.

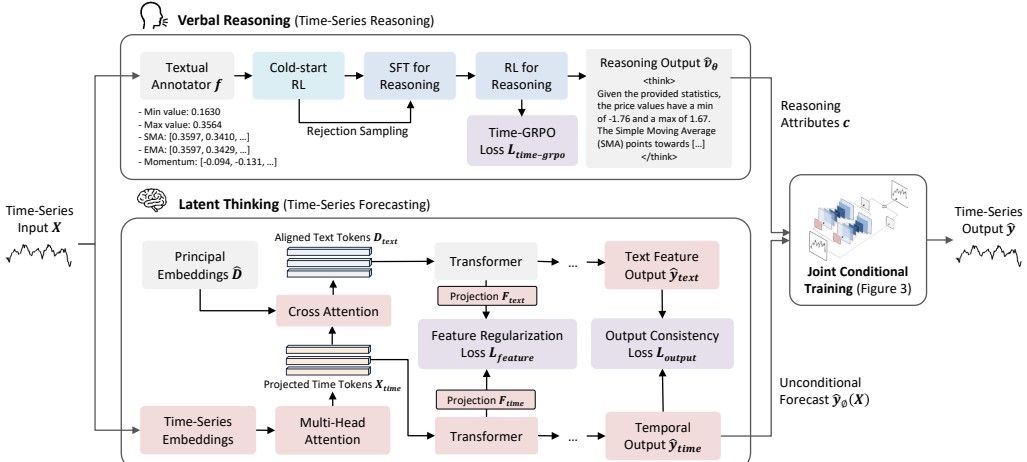

Figure 1: The Verbal Technical Analysis (VTA) framework. We first teach an LLM to reason over time-series data. The reasoning outputs are used to condition a time-series forecasting model, to produce forecasts with similar attributes. This results in forecasts with interpretable reasoning traces.

### 3.1 PROBLEM FORMULATION

We consider the task of forecasting short-term future stock prices, based on a historical window of $T$ trading days. Let $\mathbf{X} = \{\mathbf{x}_{t-T+1}, \mathbf{x}_{t-T+2}, \cdots, \mathbf{x}_t\}$, where the input vector consists of the open price, high price, low price, volume traded, closing price and adjusted closing price, i.e., $\mathbf{x}_t = [o_t, h_t, l_t, v_t, c_t, p_t]$. We aim to generate an output $\mathbf{Y} = \{\mathbf{v}, \mathbf{y}\}$, which consists of the verbal reasoning trace $\mathbf{v}$ and the price trajectory over the next $T'$ trading days $\mathbf{y} = \{p_{t+1}, p_{t+2}, \cdots, p_{t+T'}\}$.

### 3.2 TIME-SERIES REASONING

To teach an LLM to reason over time-series inputs, we use a textual annotator to extract useful interpretable signals for forecasting. The LLM uses these indicators to reason over the time-series to make forecasts without any supervision data. This is achieved through our proposed Time-Series

Group Relative Policy Optimization (Time-GRPO) method, which uses a multi-stage reinforcement learning (RL) pipeline, together with a modified GRPO objective (Shao et al., 2024).

The time-series input is first converted into textual annotations (Lin et al., 2024), which consist of its statistics information (Jin et al., 2024) (*e.g.,* its mean, minimum and maximum values) and financial technical indicators (Murphy, 1999) (*e.g.,* moving averages, momentum, *etc.*). Formally, we have:

$$\mathbf{X}' = \mathbf{f}(\mathbf{X}), \tag{1}$$

where $\mathbf{f}$ contains the annotation functions and $\mathbf{X}'$ are the annotated values. A full list of the financial technical indicators used, with their descriptions and calculations, are provided in Appendix B.

**Training Objectives.** Using both the time-series $\mathbf{X}$ and their annotations $\mathbf{X}'$, we form a prompt $\mathbf{q}$, and let the LLM forecast the upcoming time-series sequence through verbal reasoning. The objective of the LLM is to produce an output $\mathbf{o}$, which consists of a sequence prediction $\hat{\mathbf{y}}_\theta$ and a verbal reasoning trace $\hat{\mathbf{v}}_\theta$. Formally, we denote the set of all task prompts as $\mathcal{Q}$ and a group of generated outputs as $\mathbf{O} = \{\mathbf{o_1}, \mathbf{o_2}, \cdots, \mathbf{o_G}\}$. The time-series reasoning LLM policy $\pi_\theta$ is then optimized across all groups using the following Time-GRPO objective:

$$\mathcal{L}_{\text{time-grpo}}(\theta) = \mathbb{E}_{\mathbf{q} \sim \mathcal{Q}, \{\mathbf{o_i}\}_{i=1}^G \sim \pi_{\theta_{\text{old}}}(\mathbf{O}|\mathbf{q})}$$

$$\frac{1}{G} \sum_{i=1}^G \left( \min \left( \frac{\pi_\theta(\mathbf{o_i}|\mathbf{q})}{\pi_{\theta_{\text{old}}}(\mathbf{o_i}|\mathbf{q})} A_i, \text{clip} \left( \frac{\pi_\theta(\mathbf{o_i}|\mathbf{q})}{\pi_{\theta_{\text{old}}}(\mathbf{o_i}|\mathbf{q})}, 1 - \epsilon, 1 + \epsilon \right) A_i \right) - \beta \mathbb{D}_{\text{KL}}(\pi_\theta \| \pi_{\text{ref}}) \right), \tag{2}$$

$$\mathbb{D}_{\text{KL}}(\pi_\theta \| \pi_{\text{ref}}) = \frac{\pi_{\text{ref}}(\mathbf{o_i}|\mathbf{q})}{\pi_\theta(\mathbf{o_i}|\mathbf{q})} - \log \frac{\pi_{\text{ref}}(\mathbf{o_i}|\mathbf{q})}{\pi_\theta(\mathbf{o_i}|\mathbf{q})} - 1, \tag{3}$$

where $\epsilon$ and $\beta$ are hyper-parameters. $A_i$ denotes the advantage of the LLM policy, which is derived from a set of rewards $\{r_1, r_2, \cdots, r_G\}$ that are associated with outputs $\mathbf{O}$ produced in each group:

$$A_i = \frac{r_i - \text{mean}(\{r_1, r_2, \cdots, r_G\})}{\text{std}(\{r_1, r_2, \cdots, r_G\})}. \tag{3}$$

We utilize the format reward that was used in previous works (Guo et al., 2025), that enforces the model to always employ a thinking process that is between $<think>$ and $</think>$ tags.

Ideally, the generated reasoning trace should also maximize the expected accuracy of the time-series forecasts. This is achieved by utilizing the Mean-Squared Error (MSE) score as an additional reward:

$$r_{\text{MSE}}(\theta) = 1 / \left( \lambda \cdot \|\hat{\mathbf{y}}_\theta - \mathbf{y}\|_2^2 \right), \tag{4}$$

where $\lambda$ is a hyperparameter. The inverse MSE was used as the reward scores are to be maximized.

**Training Pipeline.** Following established practices in LLM fine-tuning literature (Guo et al., 2025; Ouyang et al., 2022; Lu et al., 2024; Chow et al., 2024), Time-GRPO utilizes a multi-stage pipeline to fine-tune the time-series reasoning LLM.

The first stage represents the cold-start phase (Guo et al., 2025). As there were no "gold-standard" supervision data, this stage is used for generating the initial training samples, guided by the $\mathcal{L}_{\text{time-grpo}}$ objective. Empirically, we find that the forecasting performance would not significantly improve in this stage, but the process lets us generate training data for the next stage to fine-tune the base model.

The second stage focuses on teaching the model to produce more effective reasoning. This is achieved through rejection sampling, where we keep only the reasoning traces that lead to forecasts with lower Mean Squared Error (MSE). To ensure better consistency of training samples, we also bucket the samples across different stocks and time-periods, and filter for those with MSE in the bottom $10^{\text{th}}$ percentile in each bucket. The model is then trained on these filtered samples through the use of supervised fine-tuning (SFT).

The third stage optimizes the model for the best forecasting performance for our Technical Analysis (TA) task. Given that the model has learnt to reason over time-series data in the previous stage, this stage now aims to search for the best reasoning policy that can maximize the expected accuracy of the predicted time-series. For this stage, the model is also optimized using the $\mathcal{L}_{\text{time-grpo}}$ objective.

### 3.3 TIME-SERIES FORECASTING

To perform time-series forecasting, we employ LLM-based time-series models. Works have shown that the powerful contextual modeling capabilities of LLMs can be effectively adapted for time-series forecasting tasks (Zhou et al., 2023; Chang et al., 2023; Cao et al., 2023). A key technique is to align the time-series and language distributions (Sun et al., 2023; Jin et al., 2024) such that the model is able to understand the context of time-series data (*e.g.,* up, down, steady, *etc.*). For our backbone model, we repurpose an LLM for cross-modal fine-tuning (Liu et al., 2025a).

For this step, we first pass the time-series input $\mathbf{X}$ through an embedding layer, followed by a multi-head attention layer, to obtain the projected time tokens $\mathbf{X}_{\text{time}}$. Next, it is observed that similar words are usually close to each other in the LLM embedding space, and for non-text based tasks, it is sufficient to keep cluster centers to reduce training costs (Sun et al., 2023; Liu et al., 2025a). To do this, we perform Principal Component Analysis (PCA) to retrieve the principal word embeddings $\hat{\mathbf{D}}$. Following this, we then pass the projected time tokens $\mathbf{X}_{\text{time}}$ and the principal word embeddings $\hat{\mathbf{D}}$ through a Multi-head Cross-Attention layer. This lets us align the time tokens and word embeddings in the forecasting model's embedding space to obtain the aligned cross-modal text tokens, *i.e.,*

$$\mathbf{X}_{\text{text}} = \text{Softmax}\left(\frac{\mathbf{Q}\mathbf{K}^\top}{\sqrt{C}}\right)\mathbf{V},$$

$$\text{where} \quad \mathbf{Q} = \mathbf{X}_{\text{time}}\mathbf{W}_q, \quad \mathbf{K} = \hat{\mathbf{D}}\mathbf{W}_k, \quad \mathbf{V} = \hat{\mathbf{D}}\mathbf{W}_v. \tag{5}$$

$\mathbf{W}_q$, $\mathbf{W}_k$ and $\mathbf{W}_v$ are the projection matrices for query, key and value in the multi-headed attention layer, and $C$ is the embedding dimension per attention head. $\mathbf{X}_{\text{text}}$ refers to the aligned text tokens.

Next, the projected time tokens $\mathbf{X}_{\text{time}}$ and aligned text tokens $\mathbf{X}_{\text{text}}$ are passed through consecutive LLM transformer blocks. To guide modality alignment, after each transformer block in the temporal and text branches, the embeddings pass through a projection layer (Chen et al., 2020) and are matched via a feature regularization loss. This ensures that the text representations are consistent with the temporal dynamics at each layer. Formally, given $\mathbf{F}_{\text{time}}^n$ and $\mathbf{F}_{\text{text}}^n$, which are outputs of the $n^{\text{th}}$ transformer block in the temporal and text branches, we define the feature regularization loss as:

$$\mathcal{L}_{\text{feature}} = \sum_{n=1}^{N} \gamma^{(N-n)} \, \text{sim}\big(\phi_{\text{text}}^n(\mathbf{F}_{\text{text}}^n), \phi_{\text{time}}^n(\mathbf{F}_{\text{time}}^n)\big), \tag{6}$$

where $\gamma^{(N-n)}$ are the scaling hyperparameters, $\text{sim}(\cdot, \cdot)$ is the $L_1$ regularization loss to ensure embedding similarity, and $\phi_{\text{time}}^n$, $\phi_{\text{text}}^n$ are the projection layers in the temporal and textual branches.

At the end of the transformer blocks, the features are passed through a final dense layer to produce the temporal-based and text-based outputs, $\hat{\mathbf{y}}_{\text{time}}$ and $\hat{\mathbf{y}}_{\text{text}}$. These are also matched via $L_1$ loss:

$$\mathcal{L}_{\text{output}} = \text{sim}(\hat{\mathbf{y}}_{\text{time}}, \hat{\mathbf{y}}_{\text{text}}). \tag{7}$$

The temporal-based output $\hat{\mathbf{y}}_{\text{time}}$ is extracted, which we denote as the time-series forecast $\hat{\mathbf{y}}_\phi(\mathbf{X})$.

### 3.4 JOINT CONDITIONAL TRAINING

On its own, the time-series forecasting pipeline represents a black-box model, given that the embedding space of the LLM blocks have been modified, resulting in only time-series outputs. To preserve the interpretability of the time-series forecasts, we *condition* the time-series forecasts on the outputs produced by the reasoning model. At the same time, to also preserve the forecast accuracy, we fine-tune the model to optimize for both the conditional and unconditional forecasts concurrently.

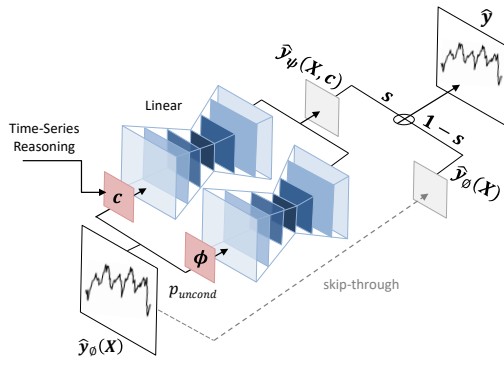

Figure 2: Joint conditional training component.

For this step, we first prompt for the reasoning output $\mathbf{o}$ using the time-series reasoning policy

$\pi_\theta$. Next, we extract descriptive attribute classes $\mathbf{c}$ (*i.e.,* its maximum, minimum, mean values) from the generated time-series $\hat{\mathbf{y}}_\theta$, which are used to condition (Dhariwal & Nichol, 2021) the time-series forecasts via joint conditional training (see Figure 2): For each label, we concatenate it with the time-series forecasts $\hat{\mathbf{y}}_\phi(\mathbf{X})$ from the time-series forecasting model. These inputs pass through separate linear layers for fine-tuning, and are then aggregated via a projection layer to generate the conditioned time-series forecasts, which we denote as $\hat{\mathbf{y}}_\psi(\mathbf{X}, \mathbf{c})$.

Finally, we use a single neural network to parameterize both the conditional and unconditional models (Ho & Salimans, 2022). Both the unconditional and conditional pipelines are trained concurrently by randomly setting $\mathbf{c}$ to the unconditional class identifier $\varnothing$ with a predefined probability $p_{\text{uncond}}$. The model is then fine-tuned using MSE loss with the ground truth series $\mathbf{y}$. We have:

$$\mathcal{L}_{\text{forecast}}(\phi) = \mathbb{E}_{\mathbf{X}, \mathbf{y}, \mathbf{c}} \left[ \|\hat{\mathbf{y}}_\psi(\mathbf{X}, \tilde{\mathbf{c}}) - \mathbf{y}\|^2 \right], \tag{8}$$

$$\tilde{\mathbf{c}} \sim \begin{cases} \mathbf{c}, & \text{with probability } 1 - p_{\text{uncond}} \\ \varnothing, & \text{with probability } p_{\text{uncond}} \end{cases}. \tag{9}$$

During inference, our forecast is then a combination of the conditional and unconditional forecasts:

$$\hat{\mathbf{y}} = s \cdot \hat{\mathbf{y}}_\psi(\mathbf{X}, \mathbf{c}) + (1 - s) \cdot \hat{\mathbf{y}}_\theta(\mathbf{X}), \tag{10}$$

where $s$ is a hyperparameter representing the guidance scale, that controls the reasoning guidance.

## 4 EXPERIMENTS

**Dataset:** We evaluate our Verbal Technical Analysis (VTA) model extensively across multiple datasets. The first is the **ACL18** StockNet dataset (Xu & Cohen, 2018), which includes historical price data for 88 U.S. stocks that are selected to represent the top 8-10 companies by market capitalization in each of the major industries. The dataset spans the period of 01/09/2012 to 01/09/2017. This dataset is a standard stock prediction benchmark that has been evaluated in multiple works (Feng et al., 2018; Sawhney et al., 2020; Feng et al., 2021; Li et al., 2023; Koa et al., 2023; Chen & Wang, 2025).

To further show the generalization ability of the model, we also collect additional stock data from across the U.S., Chinese, and European markets for testing. To ensure a bias-free selection, we choose the stocks from well-known indices, *i.e.,* the Dow Jones, the FTSE China A50 Index and the EURO STOXX 50. For these datasets, we test on the time period from 01/01/2024 to 01/01/2025.

**Baselines:** We compare against 12 state-of-the-art time-series methods: Transformer (Vaswani et al., 2017), Reformer (Kitaev et al., 2020), Informer (Zhou et al., 2021), Autoformer (Wu et al., 2021), DLinear (Zeng et al., 2023), FiLM (Zhou et al., 2022), Crossformer (Zhang & Yan, 2023), MICN (Wang et al., 2023), LightTS (Campos et al., 2023), TimesNet (Wu et al., 2022), TSMixer (Chen et al., 2023) and Non-Stationary Transformer (Liu et al., 2022). We also compare with two LLM-based time-series models: TimeLLM (Jin et al., 2024) and CALF (Liu et al., 2025a). These models are not explainable, including the two LLMs, which modify the embedding space for forecasting.

For evaluation against explainable models, we compare with reasoning LLMs: GPT-4.1 mini (OpenAI, 2025) and DeepSeek-R1 (Guo et al., 2025). To do so, we prompt these models to produce the time-series forecasts (Gruver et al., 2023; Wang et al., 2024) by reasoning on the time-series inputs.

**Implementation Details:** All LLMs used in the VTA model, including the reasoning model and the transformer blocks for time-series forecasting, are trained using Low-Rank Adaptation (LoRA) (Hu et al., 2022). Both input and output lengths $T$ and $T'$ are set to 10, which is considered short-term forecasting in time-series works (Li et al., 2022; Liu et al., 2025a). Technical Analysis is typically utilized for short-term stock trading (Schwager, 1995).

For the reasoning model, we use Qwen2.5-7B-Instruct (Team, 2024) as our base model. For the forecasting model, we use GPT-2 (Radford et al., 2019) as the base model. For hyperparameters, we set the conditional probability $p_{\text{uncond}}$ to 0.3 and the guidance scale $s$ to 0.1. More details on the experimental settings and computational resources used can be found in Appendix A.

Table 2: Performance comparison. The best baselines are underlined, and the best results are bolded.

| | StockNet | | China A50 | | EUROSTOXX 50 | | Dow Jones | | All Data | | % Improvement | |
|---|---|---|---|---|---|---|---|---|---|---|---|---|
| | MSE | MAE | MSE | MAE | MSE | MAE | MSE | MAE | MSE | MAE | MSE | MAE |
| **Large Language Models** | | | | | | | | | | | | |
| GPT-4.1 mini (OpenAI, 2025) | 0.0846 | 0.1827 | 0.4875 | 0.3191 | 0.0997 | 0.2128 | 0.1340 | 0.2358 | 0.2014 | 0.2376 | 0.4153 | 0.1072 |
| DeepSeek-R1 (Guo et al., 2025) | 0.0788 | 0.1853 | 0.2920 | 0.3093 | 0.0776 | 0.2095 | 0.1227 | 0.2251 | 0.1428 | 0.2323 | 0.1750 | 0.0868 |
| | | | | | | | | | | | | |
| **Time-Series Models** | | | | | | | | | | | | |
| Informer (Zhou et al., 2021) | 2.1846 | 0.9778 | 4.6823 | 1.1080 | 2.8986 | 1.0968 | 3.7031 | 1.1255 | 3.3672 | 1.0770 | 0.9650 | 0.8030 |
| Transformer (Vaswani et al., 2017) | 1.5071 | 0.7762 | 3.9865 | 0.9784 | 2.1002 | 0.9042 | 2.8248 | 0.9194 | 2.6047 | 0.8946 | 0.9548 | 0.7628 |
| Crossformer (Zhang & Yan, 2023) | 1.1848 | 0.6475 | 4.3396 | 0.9641 | 1.5656 | 0.7644 | 2.3503 | 0.8173 | 2.3601 | 0.7983 | 0.9501 | 0.7343 |
| TSMixer (Chen et al., 2023) | 1.4974 | 0.8193 | 3.5863 | 1.0905 | 1.1696 | 0.7227 | 2.5077 | 0.8473 | 2.1902 | 0.8700 | 0.9462 | 0.7561 |
| Reformer (Kitaev et al., 2020) | 1.1823 | 0.7628 | 2.4537 | 0.8503 | 1.5342 | 0.8302 | 2.4280 | 0.9048 | 1.8995 | 0.8370 | 0.9380 | 0.7465 |
| LightTS (Campos et al., 2023) | 0.6081 | 0.5213 | 1.0634 | 0.5509 | 0.6160 | 0.5128 | 1.0994 | 0.6103 | 0.8467 | 0.5488 | 0.8609 | 0.6134 |
| DLinear (Zeng et al., 2023) | 0.1589 | 0.3021 | 0.3880 | 0.4126 | 0.1716 | 0.3189 | 0.2407 | 0.3566 | 0.2398 | 0.3475 | 0.5088 | 0.3896 |
| FiLM (Zhou et al., 2022) | 0.0806 | 0.1927 | 0.2852 | 0.3085 | 0.0894 | 0.2157 | 0.1246 | 0.2370 | 0.1449 | 0.2385 | 0.1873 | 0.1103 |
| Non-stationary (Liu et al., 2022) | 0.0729 | 0.1822 | 0.2723 | 0.2993 | 0.0861 | 0.2079 | 0.1207 | 0.2305 | 0.1380 | 0.2300 | 0.1463 | 0.0776 |
| MICN (Wang et al., 2023) | 0.0764 | 0.1878 | 0.2498 | 0.2922 | 0.0874 | 0.2108 | 0.1373 | 0.2405 | 0.1377 | 0.2328 | 0.1449 | 0.0888 |
| Autoformer (Wu et al., 2021) | 0.0748 | 0.1866 | 0.2427 | 0.2947 | 0.0853 | 0.2103 | 0.1132 | 0.2273 | 0.1290 | 0.2297 | 0.0868 | 0.0765 |
| TimesNet (Wu et al., 2022) | 0.0708 | 0.1789 | 0.2527 | 0.2902 | 0.0796 | 0.2022 | 0.1112 | 0.2203 | 0.1286 | 0.2229 | 0.0841 | 0.0482 |
| | | | | | | | | | | | | |
| **Time-Series LLMs** | | | | | | | | | | | | |
| TimeLLM (Jin et al., 2024) | 0.0704 | 0.1780 | 0.2439 | 0.2850 | 0.0776 | 0.1993 | 0.1127 | 0.2216 | 0.1262 | 0.2210 | 0.0663 | 0.0400 |
| CALF (Liu et al., 2025a) | 0.0674 | 0.1738 | 0.2412 | 0.2843 | 0.0762 | 0.1957 | 0.1092 | 0.2181 | 0.1235 | 0.2180 | 0.0460 | 0.0267 |
| | | | | | | | | | | | | |
| **Our Model** | | | | | | | | | | | Avg | Avg |
| VTA (Ours) | **0.0659** | **0.1701** | **0.2265** | **0.2737** | **0.0748** | **0.1929** | **0.1040** | **0.2120** | **0.1178** | **0.2122** | 0.4672 | 0.3417 |

## 5 RESULTS

**Performance Comparison.** Table 2 reports the forecasting results. We can observe the following:

- The inference-only reasoning LLMs (*i.e.,* GPT-4.1 mini, DeepSeek-R1) do not show very strong performances, as they are likely not fine-tuned for time-series forecasting. However, they were still able to beat some of the fine-tuned time-series models (*e.g.,* Transformer, DLinear), which demonstrate some effectiveness of verbally reasoning over time-series inputs to do forecasting.

- Among the time-series baselines, the biggest performance jump comes from the models which decomposes the data by trend and seasonality (*i.e.,* FiLM, MICN, Autoformer and TimesNet). This could be attributed to the characteristics of stock prices, which are often affected by long-term and short-term business cycles (Dalio, 2018). Another notable model that performs well is the non-stationary transformer, which might be attributed to the non-stationary behavior of stock price data (Malkiel, 2019).

- The time-series LLMs are able to surpass the performance of non-LLM-based models. These models typically align the LLM's internal word embeddings to time-series embeddings to do time-series forecasting. It is possible that the intrinsic knowledge of the LLM helps the model to understand the characteristics of stock data, thus allowing it to capture forecasting patterns better.

- Our proposed VTA model demonstrates the best performance in stock forecasting in both MSE and MAE. VTA explicitly combines internal (latent) understanding with external (verbalized) reasoning. The empirical improvements suggest that integrating these two techniques can be beneficial for time-series forecasting. In addition to improved accuracy, the VTA model also produces interpretable reasoning traces for its forecasts, which do not exist in most baseline models.

More details of the statistical significance of the experiments can be found in Appendix C.

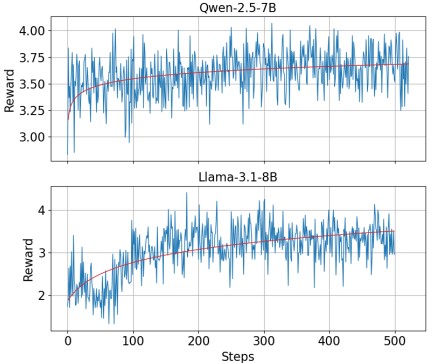

Figure 3: Correctness reward over steps.

Table 3: Ablation study of the LLM fine-tuning stages.

| | | Llama-3.1-8B | Qwen-2.5-3B | Qwen-2.5-7B (Ours) |
|---|---|---|---|---|
| Base Model | MSE | 0.1482 | 0.1707 | 0.0949 |
| | MAE | 0.2543 | 0.2181 | 0.2040 |
| Cold Start RL | MSE | 0.1475 | 0.1648 | 0.0941 |
| | MAE | 0.2536 | 0.2153 | 0.2036 |
| SFT for Reasoning | MSE | 0.1168 | 0.1032 | 0.0893 |
| | MAE | 0.2267 | 0.1884 | 0.1997 |
| RL for Reasoning | MSE | 0.0955 | 0.0832 | 0.0686 |
| | MAE | 0.2062 | 0.1745 | 0.1741 |
| Conditioning (VTA) | MSE | 0.0667 | 0.0672 | **0.0659** |
| | MAE | 0.1713 | 0.1710 | **0.1701** |

**Ablation Study.** We conduct an ablation study to demonstrate the effectiveness of the model design.

- From Figure 3, we observe that the inverse MSE reward $r_{\text{MSE}}$, a component of the Time-GRPO objective, increases across the number of training steps. This suggests that it is *possible* to learn verbal reasoning steps for time-series forecasting, which our VTA model was able to achieve.

- From Table 3, we see that each fine-tuning stage helps to improve the results of the model. However, the first RL fine-tuning, which uses the Time-GRPO objective, was not efficient by itself, showing a small improvement of 1.6% over the base model in MSE averaged across all variants.

- However, after rejection sampling and SFT to teach the model how to reason over time-series, the second RL fine-tuning, which uses the same Time-GRPO objective, produces an average improvement of 20.3%. This highlights the usefulness of fine-tuning over a multi-stage pipeline.

- Finally, conditioning on the additional forecasting model helped to improve the performance further, showing the benefit of enhancing external verbal reasoning with internal latent understanding.

**Contribution of Reasoning Component.** To study the contribution of the reasoning component on forecasting performance, we artificially corrupt the reasoning trace to observe their impact on the final forecasts. This was done in two ways: **(1) Adversarial:** We add adversarial noise to the technical indicator values; **(2) Remove Information:** Within the prompt, we force the LLM to not utilize certain indicators in its reasoning. The forecasting performances are reported in Figure 4.

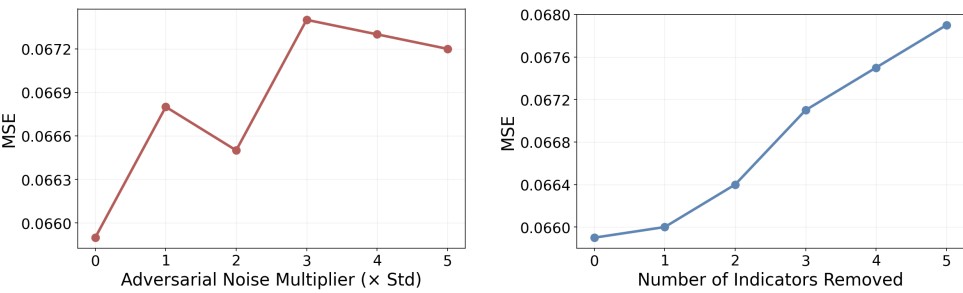

Figure 4: Change in reasoning performance when the reasoning traces are corrupted.

In Figure 4, removing the indicators leads to a clear degradation in performance, demonstrating that the reasoning traces provide genuinely useful guidance to the model. Adding adversarial noise also reduces overall performance but does not yield a consistent trend. A possible explanation is that, during joint conditional training, the model gradually learns to rely more heavily on the time-series forecasting component once it detects that the reasoning signals have become unreliable.

**Reasoning Quality.** To evaluate the quality of the reasoning traces, we refer to past works on LLM explainability (Koa et al., 2024; Lin et al., 2024) to design a set of relevant metrics for our task. Each reasoning trace and its associated time-series forecast was scored on a scale from 1 (poor) to 5 (excellent). The criteria are defined as follows:

- **Clarity**: How clearly and succinctly the reasoning explains its analysis in a structured manner.

- **Depth**: How well the reasoning incorporates explicit financial or technical indicators (*e.g.,* MACD, RSI, Bollinger Bands, EMA) to meaningfully support its conclusions.

- **Accuracy**: How precisely financial indicators are interpreted and technically described.

- **Coherence**: How logically consistent and organized the reasoning is, ensuring clear alignment between analysis and conclusion.

- **Relevance**: How directly and effectively the chosen indicators are linked to the stock-price forecast provided.

Using these metrics, we surveyed 25 industry experts with professional experience in financial market analysis, with backgrounds from organizations such as JPMorgan, UBS, Evercore, and Allianz Global Investors, *etc.* For each, we presented the model outputs from VTA (ours), GPT-4.1 mini, and Deepseek-R1, showing both the forecasts and the textual reasoning. Respondents were blind to which model produced which output and were shown 15 randomly selected samples for evaluation.

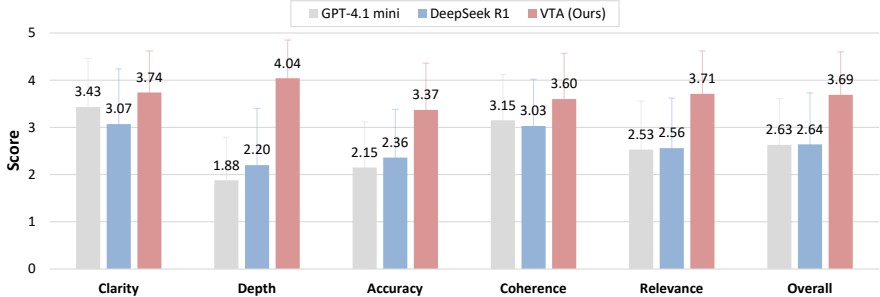

Figure 5: Performance of VTA on the financial time-series reasoning task.

From Figure 5, the results show that VTA achieved the highest average rating across all five metrics when compared to the other two LLMs. The best performance gains were observed in *Depth*, *Accuracy*, and *Relevance*. These criteria most directly reflect technical reasoning ability and the use of financial indicators. These findings suggest that our model, which was designed specifically for technical analysis, was able to produce reasoning outputs that were preferred by domain experts, demonstrating its practical strengths. The differences in *Coherence* and *Clarity* were smaller, which can be attributed to the fact that general-purpose LLMs like GPT-4.1 mini and Deepseek-R1 are good at producing fluent, well-structured text, even if they lack domain specialization. More details on this, including statistical significance and open-ended responses, are found in Appendix D.1.

**Case Studies.** To illustrate the capabilities of VTA, we present some qualitative case studies to demonstrate its reasoning process. For the figures, the horizontal axis shows the time-steps while the vertical axis shows the scaled prices. In Figure 6, we see that VTA was able to correctly reason about the (downward) price correction and subsequent upward trend. In Figure 7, VTA correctly reason about the oscillating prices with a slight uptrend. More case studies can be found in Appendix E.

<think>
The stock prices have been fluctuating significantly, with a recent upward trend. The moving averages suggest a decreasing trend, while the momentum and RSI indicate a weak buying pressure. The MACD line is slightly positive, but the Williams %R and CCI suggest overbought conditions. Bollinger Bands show the current price is within the lower band, indicating a possible bounce. However, the stochastic indicators are in the overbought region, which might lead to a correction. Given the recent trend and indicators, it's likely the stock will correct and then continue the upward trend.

Predicting the next 10 days, I'll assume a slight correction followed by a rebound, staying within the recent range.
</think>

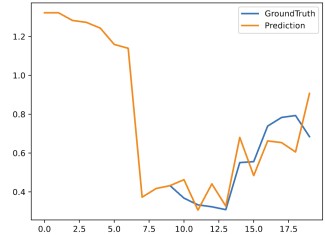

Figure 6: An example of VTA reasoning about a slight correction and possible rebound in price.

<think>
The stock price has been fluctuating but seems to be in a range between 1.349 and 1.483. The moving averages suggest a slight downward trend, while the momentum and RSI indicate that the stock is not in an overbought or oversold condition. The MACD line shows a slight positive crossover, which could indicate a potential buying opportunity. The Bollinger Bands suggest that the stock price is near the lower band, which could indicate a potential bounce. The Stochastic indicators are in a neutral range, not indicating a strong trend in either direction.

Given the recent price action and the technical indicators, the stock might continue to oscillate within the current range or slightly increase. However, the positive MACD crossover and the proximity to the lower Bollinger Band suggest a slight upward bias. I will predict a range slightly above the recent lows but not exceeding the recent highs.
</think>

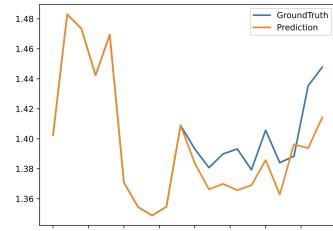

Figure 7: An example of VTA reasoning about prices oscillating within a range with a slight uptrend.

**Generalization to Other Domains.** In general, VTA is able to produce reasoning traces for any time-series data. To verify this, we run VTA on datasets from other domains. These include Healthcare (Wu et al., 2021) and Energy (Zhou et al., 2021), which contains time-series data on ILI (influenza-like illness) cases and oil temperature respectively. The generated reasoning traces are shown in Figure 8. In these domains, we observe that the time-series do not contain any intrinsic interpretable signals to reason over, and VTA do not go beyond simple trend extrapolation in its reasoning. It has also been shown in previous studies that large complex models do not meaningfully improve performance in these use cases (Tan et al., 2024). For these cases, VTA are still able to produce reasoning traces, but these do not contribute significant additional signals for the forecasting model. The forecasting performance of VTA on these datasets can be found in Appendix G.

**Healthcare (ILI dataset):**

> The data shows a clear pattern of fluctuation, with a general trend of decreasing % Weighted ILI values over the last few weeks. The most recent values are in the negative range, indicating a decline in influenza-like illness. Given this trend, it's reasonable to predict a continuation of this decline, but with some variability. The last few weeks have shown a steady decline, so we can expect this trend to continue, but with some fluctuation. I will predict a gradual decrease with some variability.

**Energy (ETTh1 dataset):**

> The given data is the standardized transformer measurements (OT) for 96 hours. The task is to predict the oil temperature for the next 24 hours. Given the nature of the data and the task, a simple moving average or a trend extrapolation can be used for prediction. However, for this example, I will use a basic approach of taking the last value of the OT series as a reference and slightly adjusting it based on the recent trend. The last value of the OT series is -0.954. The recent trend shows a slight decrease in the last few values. I will predict a slight decrease in the next 24 hours, considering the recent trend and the range of the data.

Figure 8: Examples of VTA reasoning traces on the Healthcare and Energy domains.

**Portfolio Optimization.** To justify the practical capability of VTA, we also evaluate the model in a real-life investment setting. Given that the model produces multi-step predictions, we are able to form portfolios that maximize the returns and minimize the volatility across the prediction length (Koa et al., 2023). This is achieved by performing Markowitz optimization (Markowitz, 1952) across the 10-day predictions. The portfolio is rebalanced daily, using the predicted returns and their covariance matrix.

For evaluation, we compare against similar portfolios formed using the top-5 performing time-series models and all LLM baselines. The portfolio are compared on common investment metrics, such as their Sharpe ratio (Sharpe, 1994). The explanations of the portfolio metrics used are as follows:

- **Returns**: Measure the percentage change in portfolio value over a given period, indicating overall profitability.
- **Volatility**: Captures the dispersion of returns over time, with higher volatility reflecting greater fluctuations and uncertainty.
- **Maximum Drawdown**: Represents the largest peak-to-trough decline in portfolio value, highlighting the worst observed loss during the evaluation window.
- **Sharpe Ratio**: Assesses risk-adjusted performance by comparing excess returns to return volatility, where higher values indicate more efficient risk-taking.

We evaluate the portfolios across all 4 datasets, and report the average results across each metric.

From Table 4, we see that the portfolio constructed using VTA predictions demonstrates strong overall performance. It ranks as the strongest baseline on the returns, volatility and maximum drawdown metrics, and the values are very close to the top-performing models for each. Notably, VTA achieves the highest Sharpe ratio among all models, which represents the risk-adjusted returns. Given that the Sharpe ratio is one of the most common measure of investment performance, this highlights the practical effectiveness of the VTA forecasts. More advanced financial analysis on the portfolios can be found in Appendix H.

Table 4: Comparison on common portfolio metrics.

|  | Returns | Volatility | Drawdown | Sharpe |
|---|---|---|---|---|
| GPT-4.1 mini | 0.1868 | 0.1226 | -0.0947 | 1.3096 |
| Deepseek-R1 | 0.2069 | 0.1356 | -0.1243 | 1.4074 |
| FiLM | 0.2211 | **0.1184** | -0.1085 | 1.4421 |
| Non-stationary | 0.2122 | 0.1186 | **-0.0825** | 1.4430 |
| MICN | 0.1603 | 0.1193 | -0.1094 | 1.1809 |
| Autoformer | **0.2495** | 0.1341 | -0.1121 | 1.4736 |
| TimesNet | 0.1714 | 0.1198 | -0.0947 | 1.2748 |
| TimeLLM | 0.2185 | 0.1193 | -0.1040 | 1.5230 |
| CALF | 0.2019 | 0.1247 | -0.0981 | 1.4566 |
| VTA (Ours) | 0.2409 | 0.1185 | -0.0883 | **1.7190** |

## 6  CONCLUSION

In this work, we tackled the task of verbally reasoning over financial time-series data. This task is challenging as it switches between the time-series and natural language domain for the stock price data and the reasoning step. To deal with this, we introduce our Verbal Technical Analysis (VTA) framework, which combines verbal and latent reasoning to produce interpretable time-series forecasts. The framework utilizes our Time-GRPO method to finetune the reasoning model, and conditions its forecasts on the reasoning attributes. We conducted extensive experiments and find VTA can achieve state-of-the-art forecasting accuracy while producing high-quality reasoning traces.

From the observations in this work, we propose two possible future directions. The first is to incorporate more stock characteristics into the VTA model (*e.g.,* trend and seasonality, non-stationary behavior). The second is to improve the alignment between reasoning and forecasting, possibly through the use of more advanced techniques from image conditioning control (Zhang et al., 2023).

## 7 ACKNOWLEDGEMENTS

This research is supported by the Asian Institute of Digital Finance (AIDF) and the NExT Research Centre. This research is also supported by the Singapore Ministry of Education (MOE) Academic Research Fund (AcRF) Tier 1 grant.

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

# Appendix

## Table of Contents

## A   ADDITIONAL EXPERIMENT DETAILS

### A.1   MODEL AND TRAINING HYPERPARAMETERS

This section summarizes the key hyperparameters used for training the Verbal Technical Analysis (VTA) model. All Large Language Model (LLM) components were trained using the Unsloth framework[1], which supports 4-bit quantization and Low-Rank Adaptation (LoRA) (Hu et al., 2022). The implementation of Group Relative Preference Optimization (GRPO) follows the principles outlined by the Hugging Face TRL library[2].

**Time-Series Reasoning.** The reasoning model was developed from Qwen2.5-7B-Instruct using a multi-stage training pipeline consisting of GRPO and supervised fine-tuning (SFT). The maximum sequence length was controlled via the `max_seq_length` parameter, and inference was performed with a temperature of 0.2.

LoRA was applied with a specific `lora_rank`, targeting key modules such as the attention projections (`q_proj`, `k_proj`, `v_proj`, `o_proj`) and feed-forward layers (`gate_proj`, `up_proj`, `down_proj`). During the initial and final GRPO training phases—used for format learning—a learning rate of $5 \times 10^{-6}$ was used over two epochs. Each device processed a batch size of 4, with two gradient accumulation steps. For GRPO, four generations were produced per prompt, and the combined prompt and completion length was capped at 500 tokens. Rewards were based on adherence to the target format and accuracy of the prediction.

---

[1] `https://unsloth.ai/blog/r1-reasoning`
[2] `https://huggingface.co/docs/trl/main/en/grpo_trainer`

For SFT data generation, a rejection sampling mechanism was employed to select the top 10% Mean Squared Error (MSE) examples from a total of 100 buckets. This was followed by reasoning enhancement through SFT, using a learning rate of $2 \times 10^{-4}$ over two epochs. Here, the per-device batch size was reduced to 1, while increasing the number of gradient accumulation steps to 4.

**Time-Series Forecasting.** The forecasting component of the VTA model is adapted from GPT-2 and uses a fixed input and output sequence length of 10 days (denoted as $T$ and $T'$). This component was trained for 20 epochs with a learning rate of $1 \times 10^{-4}$ and a batch size of 16. The architecture comprises 6 GPT-style transformer layers, as defined by the `gpt_layers` parameter.

LoRA was configured with a rank of 8, a scaling factor (`lora_alpha`) of 32, and a dropout rate of 0.1. The joint conditional training was implemented with a probability of unconditional training $p_{\text{uncond}} = 0.30$, and a guidance scale of $s = 0.1$. Additionally, alignment loss was incorporated using the statistical properties (*i.e.,* minimum, maximum, and mean) of the predicted sequences.

## A.2 COMPUTATIONAL RESOURCES

All experiments were conducted using $4\times$ NVIDIA A5000 GPUs (24 GB VRAM each). Full reasoning model training—including cold-start reinforcement learning, supervised fine-tuning, and reward-guided GRPO—takes approximately 120 GPU-hours for LLaMA 3.1–8B, 100 GPU-hours for Qwen 2.5–7B, and 60 GPU-hours for Qwen 2.5–3B, using Unsloth's LoRA fine-tuning implementation with `gpu_memory_utilization=0.5` (requiring around 21 GB VRAM per process).

Once reasoning traces are generated, downstream forecasting runs are significantly more efficient. Training a forecasting model for a single stock with approximately 1000 training and 250 test points takes around 3 minutes on a single GPU (using $\sim$2.7 GB VRAM). A complete run over the full StockNet dataset requires approximately 4.5 hours on one GPU.

## B LIST OF TECHNICAL INDICATORS

Table 5: A list of the financial technical indicators used in Time-GRPO.

| Indicator | Description | Formula |
|---|---|---|
| Simple Moving Average (SMA) | Identifies trend by smoothing price data over a period | $\text{SMA} = \frac{1}{n}\sum_{i=1}^{n}\text{Price}_i$ |
| Exponential Moving Average (EMA) | Measures the trend by smoothing price data with greater weight to recent prices | $\text{EMA}_t = \text{Price}_t \cdot \alpha + \text{EMA}_{t-1} \cdot (1 - \alpha)$ |
| Momentum | Tracks the speed of price changes for trend momentum | $\text{Momentum} = \text{Close}_t - \text{Close}_{t-n}$ |
| Relative Strength Index (RSI) | Identifies overbought/oversold conditions using average gains and losses | $\text{RSI} = 100 - \left(\frac{100}{1+\frac{\text{Avg Gain}}{\text{Avg Loss}}}\right)$ |
| MACD Line | Measures momentum via difference between short and long EMAs | $\text{MACD} = \text{EMA}_{12} - \text{EMA}_{26}$ |
| Williams %R | Measures overbought/oversold conditions by comparing close to recent highs | $\text{Williams } \%R = \frac{\text{Highest High}_n - \text{Close}_t}{\text{Highest High}_n - \text{Lowest Low}_n} \times (-100)$ |
| Commodity Channel Index (CCI) | Identifies price deviations from a moving average for cyclical trends | $\text{CCI} = \frac{\text{Price} - \text{MA}}{0.015 \times \text{Mean Deviation}}$ |
| Average Directional Index (ADX) | Measures trend strength using directional movement | $\text{ADX} = 100 \times \frac{\text{EMA}(|\text{DM}^+ - \text{DM}^-|)}{\text{DM}^+ + \text{DM}^-}$ |
| Bollinger Bands | Measures volatility using standard deviations around a moving average | $\text{Upper Band} = \text{MA} + k \cdot \sigma$ 
 $\text{Lower Band} = \text{MA} - k \cdot \sigma$ |
| Stochastic Oscillator | Measures momentum by comparing current close to a range of highs and lows | $\%K = \frac{\text{Close}_t - \text{Lowest Low}_n}{\text{Highest High}_n - \text{Lowest Low}_n} \times 100$ |

## C STATISTICAL SIGNIFICANCE OF MAIN RESULTS

To assess whether the differences relative to the second-best model in the model performance experiments ($\text{MSE}_0 = 0.06737$, $\text{MAE}_0 = 0.17380$) are statistically significant, we ran our model $n = 10$ times with different random seeds and performed one-sample, one-sided Student's $t$-tests under the

alternative hypothesis that our model's errors are lower than those of the second-best model. Let $\bar{x}$ and $s$ denote the sample mean and sample standard deviation of the metric over the 10 runs.

$$t = \frac{\bar{x} - \mu_0}{s/\sqrt{n}},$$

where $\mu_0$ is the error of the second-best model. We reject the null hypothesis for $p < 0.05$. Both

Table 6: One-sample, one-sided $t$-test results against the second-best model.

| Metric | $\bar{x}$ | $s$ | $t$ | $p$ (one-sided) |
|--------|-----------|-----|-----|-----------------|
| MSE | 0.06661 | 0.00054 | $-4.42$ | $8.3 \times 10^{-4}$ |
| MAE | 0.17081 | 0.00047 | $-20.04$ | $4.5 \times 10^{-9}$ |

tests yield $p \ll 0.05$, indicating that the reductions in error relative to the second-best model are statistically significant. For completeness, we report the full distribution over seeds as:

$$\text{MSE} = 0.06661 \pm 0.00054, \quad \text{MAE} = 0.17081 \pm 0.00047 \quad (\text{mean} \pm \text{std}, n = 10).$$

We select the single best run (MSE = 0.06594, MAE = 0.17008) for all subsequent evaluations.

## D    DETAILS ON EXPERT EVALUATION

### D.1    SUPPLEMENTARY INFORMATION

We conducted significance tests for our expert evaluation results:

Table 7: Significance testing (paired $t$-tests). Values indicate $p$-values for pairwise comparisons.

| Comparison | Clarity | Depth | Accuracy | Coherence | Relevance | Overall |
|------------|---------|-------|----------|-----------|-----------|---------|
| VTA vs GPT-4.1 mini | 0.0765 | $9.1 \times 10^{-10}$ | $4.4 \times 10^{-6}$ | 0.0280 | $1.6 \times 10^{-5}$ | $1.3 \times 10^{-6}$ |
| VTA vs Deepseek R1 | 0.0024 | $1.8 \times 10^{-7}$ | $3.1 \times 10^{-5}$ | 0.0034 | $1.0 \times 10^{-5}$ | $2.3 \times 10^{-6}$ |
| GPT-4.1 mini vs Deepseek R1 | 0.0555 | 0.0045 | 0.0247 | 0.4594 | 0.6854 | 0.8515 |

A statistical analysis using paired $t$-tests shows that the differences between VTA and the other two models are significant at the 5% level for all criteria except *Clarity*, where the base LLMs are already good at. Thus, the higher expert ratings for our model's *reasoning* are statistically robust.

In addition to the quantitative scores, we also allowed the experts to provide open-ended responses on the strengths and weaknesses of the models. We summarize the key points here:

Experts highlighted the strengths of VTA in using a wider variety of relevant indicators and in providing conclusions that align with the explanation (*e.g., "The analysis included a variety of different indicators and used them to create a coherent story"*, *"Interesting use and relevance of indicators"*). In contrast, outputs from Deepseek-R1 and GPT-4.1 mini tended to be either vague in analysis depth or not clearly linked to the price forecast, especially in the case of Deepseek-R1.

On the other hand, experts suggested that VTA could further improve by discussing indicators in more detail and by using clearer formatting (*e.g., "Should use bullet points for readability"*, *"Should specify explicit indicator thresholds"*). While better formatting and longer explanations can be addressed in future work, precise thresholds could be more difficult to enforce since the model learns them adaptively from data.

### D.2    SURVEY PARTICIPANTS

For the expert evaluation, we have surveyed domain experts with professional experience in financial market analysis. A list of their surveyed background are as follows (duplicates are removed):

*J.P. Morgan, UBS, Allianz Global Investors, Evercore, Perella Weinberg Partners, Bain & Company, McKinsey & Company, L.E.K. Consulting, H.I.G. Capital, Simon Kucher, Axxion S.A., TCG Corporate Finance, BayernLB, Spinone Capital, Phillip Nova, Check24, SD Guthrie, MSMIF, Asian Institute of Digital Finance, Harvard Business School*

# E  ADDITIONAL CASE STUDIES

<think>
The stock prices have been fluctuating but generally increasing over the past 10 days. The moving averages suggest a slight upward trend, but the momentum and RSI indicate a potential reversal or pullback. The MACD line is slightly negative, suggesting a bearish signal. The Williams %R and CCI show that the stock is overbought, which could lead to a correction. Bollinger Bands show the stock is close to the lower band, indicating support. The stochastic indicators are in the overbought region, which might lead to a pullback.

Given the recent trend and indicators, it's likely that the stock will experience some volatility but may not see a significant increase in the next 10 days. A pullback is possible, but the support from the Bollinger Bands and the recent upward trend suggest that the stock might not fall too much.

Predicting the exact prices is challenging, but based on the current trend and indicators, a moderate increase with some pullbacks is likely.
</think>

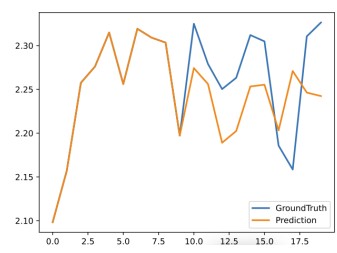

Figure 9: VTA reasoning about a moderate increase (from the last price) with some pullbacks.

<think>
The stock price has been fluctuating but generally trending upwards over the past 10 days. The moving averages suggest a slight upward trend, with the exponential moving average showing a more recent upward trend. The momentum indicator has been positive, indicating an upward momentum. The RSI is above 50, suggesting the stock is not oversold. The MACD line has been positive, indicating a bullish trend. The Williams %R is in a relatively low range, suggesting the stock is not overbought. The Bollinger Bands show the stock price is within the upper band, which could indicate a potential reversal. The stochastic indicators are also in a positive range, suggesting the stock is in an uptrend.

Given the current positive momentum and trend, and the fact that the stock price is near the upper Bollinger Band, it might be expected to correct or consolidate in the near future. However, the overall trend remains positive. A reasonable prediction would be a slight correction followed by a continuation of the upward trend.

Let's predict the next 10 days' closing prices to be slightly lower than the current price but still within the recent range.
</think>

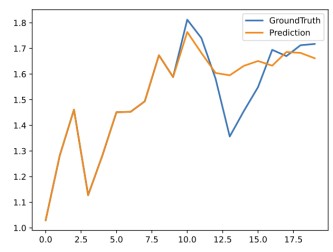

Figure 10: VTA reasoning about prices to be slightly lower than last price but within recent range.

<think>
The stock has been showing a trend of recovery from its minimum price of -0.7191 to its maximum price of -0.1486. The moving averages suggest a slight upward trend, with the simple moving average (SMA) and exponential moving average (EMA) both indicating positive values. The momentum and relative strength index (RSI) are also positive, indicating a bullish trend. The MACD line is positive, suggesting a bullish momentum. The Williams %R and Commodity Channel Index (CCI) suggest that the stock is not overbought and is in a bullish phase. The Bollinger Bands show that the stock price is within the upper band, indicating a strong bullish trend. The stochastic indicators are also positive, suggesting that the stock is in an uptrend.

Given these indicators, it is likely that the stock will continue to rise in the next 10 days. However, the exact values are hard to predict precisely. We can assume a slight increase in the closing prices, staying within the range of the current prices and the upper Bollinger Band.
</think>

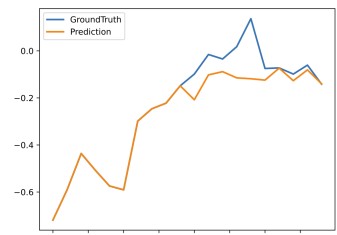

Figure 11: VTA reasoning about a slight increase in price but staying within the current range.

<think>
The stock has been showing a downward trend with the lowest price on day 9 and the highest on day 3. The moving averages suggest a declining trend as well. The RSI is quite low, indicating the stock might be oversold. The MACD line is also negative, suggesting a bearish trend. The Williams %R is very negative, indicating strong overselling. The Bollinger Bands show the stock price is below the middle band, which is consistent with a bearish trend. The Stochastic Oscillator is also in the oversold region, which might suggest a potential bounce.

Given the current trend and indicators, it's likely that the stock will continue to decline but might experience a short-term bounce. However, the overall trend is bearish. I will predict a slight decline with a potential short-term bounce.
</think>

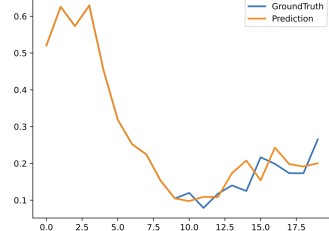

Figure 12: VTA reasoning about a potential short-term bounce, which realized in the ground truth.

<think>
The stock prices have been increasing over the past 10 days, with a recent peak on day 9. The moving averages suggest a slight upward trend, but the momentum and Williams %R indicate some overbought conditions. The Bollinger Bands show the current price is above the middle band, but not yet at the upper band, suggesting a potential for a pullback. However, the RSI and MACD are still in positive territory, indicating a bullish trend. Given the recent increase and the positive indicators, the stock is likely to continue its upward trend, but with some potential for a pullback.

Predicting the exact prices is challenging, but we can use the recent trend and indicators to make an educated guess. The prices are likely to remain above the recent lows but not necessarily break new highs. We can assume a slight increase from the last price, with some potential for a pullback.
</think>

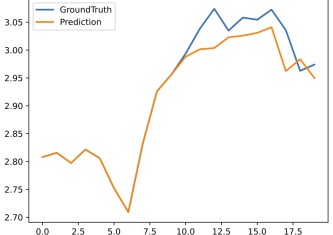

Figure 13: VTA reasoning about a slight increase, with potential for a pullback (*i.e.,* trend reversal).

## F  LLM-AS-A-JUDGE

To ensure reproducibility of the evaluation method, we further evaluated the quality of the reasoning generated by VTA using LLM-as-a-judge (Zheng et al., 2023; Gu et al., 2024). To do this, we first randomly sample 1,000 reasoning traces for evaluation. The reasoning samples are then evaluated using a stronger model as the judge, GPT-4.1 (OpenAI, 2025). The samples are judged on the same metrics on a scale of 1-5 and the average scores over all samples is reported.

Table 8: Comparison on reasoning quality across different VTA variants and reasoning LLMs.

|  | Clarity | Depth | Accuracy | Coherence | Relevance | Overall |
|---|---|---|---|---|---|---|
| GPT-4.1 mini | 4.06 | 1.87 | 2.37 | 4.05 | 2.29 | 2.93 |
| Deepseek-R1 | 3.95 | 3.02 | 2.97 | 3.98 | 3.25 | 3.43 |
| VTA (Ours) | **4.21** | **4.14** | **3.96** | **4.57** | **4.58** | **4.29** |

From Table 8, we observe that there is a clear improvement in all metrics from the inference-only reasoning models to our VTA model, showing the effectiveness of the fine-tuning process. Importantly, when comparing these LLM-as-judge results to the human experts ratings, we also find that the relative differences are highly consistent: VTA shows the largest margin over baselines in Depth, Accuracy, and Relevance, while the smallest gap is seen in Coherence and Clarity.

## G  COMPARISON IN OTHER DOMAINS

Table 9 reports the performance comparison on the Healthcare (ILI) and Energy (ETTh1) datasets.

Table 9: Performance comparison on time-series from other domains.

|  | ILI | | ETTh1 | |
|---|---|---|---|---|
|  | MSE | MAE | MSE | MAE |
| Informer | 1.7372 | 0.8669 | 0.3128 | 0.4848 |
| Transformer | 1.6827 | 0.8122 | 0.2505 | 0.4360 |
| Crossformer | 1.3320 | 0.7270 | 0.1298 | 0.2914 |
| TSMixer | 2.1163 | 0.9504 | 0.0877 | 0.2375 |
| Reformer | 1.5554 | 0.8134 | 0.2422 | 0.4020 |
| LightTS | 2.3121 | 1.0612 | 0.0625 | 0.1921 |
| DLinear | 2.9160 | 1.3933 | 0.0404 | 0.1499 |
| FiLM | 2.6761 | 1.3856 | 0.0423 | 0.1543 |
| Non-stationary | 1.4328 | 0.8538 | 0.0403 | 0.1530 |
| MICN | 1.3431 | 0.8732 | 0.3063 | 0.4649 |
| Autoformer | 1.7554 | 1.0350 | 0.0736 | 0.2146 |
| TimesNet | **1.0795** | **0.6910** | 0.0410 | 0.1572 |
| CALF | 1.4442 | 0.7409 | 0.0361 | 0.1411 |
| VTA (Ours) | 1.4366 | 0.7090 | **0.0346** | **0.1374** |

Here, we observe that the healthcare and energy datasets do not follow the same performance patterns we saw in financial time-series, which could be attributed to different defining characteristics. For example, models that benefit from trend-seasonal decomposition on financial data do not show the same advantage here, likely because ILI and ETTh1 do not exhibit the same cyclical structure.

In these domains, especially on the ILI dataset, our VTA model also does not exhibit performance that is too far away from the time-series LLM model (CALF). It is possible that the additional reasoning traces provide limited benefit when there is not much complex signals to reason over outside of simple trend extrapolation, which was previously visualized in Figure 8.

## H  FINANCIAL ANALYSIS OF PORTFOLIOS

To assess the robustness of our portfolio returns, we apply standard performance attribution models widely used in finance. These models allow us to separate returns that can be explained by common risk exposures from those that reflect potential strategy-specific value.

- **Capital Asset Pricing Model (CAPM)**: A baseline model that relates portfolio returns to overall market returns. It provides a measure of whether the strategy delivers excess returns (alpha) after adjusting for market risk (beta).
- **Fama–French multi-factor models**: Extensions of CAPM that incorporate additional risk factors, such as company size, value versus growth, profitability, investment patterns, and momentum. These factors capture well-documented drivers of returns beyond market exposure.

These models enables us to evaluate whether our results are explained by general market and factor exposures or whether they demonstrate incremental performance beyond established benchmarks.

Following industry practice, we conduct an in-depth performance attribution study of the portfolio formed from our VTA method, using both CAPM and the Fama-French 6-factor model.

### H.1  THE CAPM REGRESSION MODEL

For CAPM, the following model was applied to the daily returns of our portfolio and the market:

$$(R_{\text{VTA},t} - R_{f,t}) = \alpha + \beta \cdot (R_{\text{market},t} - R_{f,t}) + \epsilon_t,$$

where:

- $(R_{\text{portfolio},t} - R_{f,t})$: The excess return of our portfolio on day $t$.
- $(R_{\text{market},t} - R_{f,t})$: The excess return of the market benchmark on day $t$.
- **Alpha** ($\alpha$): The regression intercept, which represents the portion of the portfolio's return that is not explained by market movements.
- **Beta** ($\beta$): The regression slope, which measures the portfolio's systematic risk relative to the market.
- **Epsilon** ($\epsilon_t$): The error term for day $t$.

A positive and statistically significant alpha is indicative of a superior strategy, whereas an alpha of 0 would mean that it has the same performance as the benchmark CAPM method. Additionally, we also report the R-squared of the methods against the benchmark, which show how much of the return variation can be explained by CAPM. For investors, low R-squared would be ideal as they show that the trading signals are less correlated, which reduces the idiosyncratic risk of the portfolio.

We regressed our daily portfolio returns against the excess market return for each region, using the representative market indices. Below is a summary of results across the four datasets:

The CAPM regression results provide a useful first diagnostic for understanding the risk-adjusted performance of our strategy. Across all four datasets, the strategy exhibits positive annualized alpha, which suggests that returns exceed those predicted by exposure to the overall market alone. However, statistical significance is only achieved in the China A50 dataset, where the alpha is both high

Table 10: CAPM Regression Results.

| Dataset | Market Index Used | Beta ($\beta$) | Annualized Alpha ($\alpha$) | Alpha p-value | R-squared |
|---|---|---|---|---|---|
| dowjones_30 | Dow Jones Industrial (DJI) | 0.6745 | +3.41% | 0.701 | 44.04% |
| stocknet | MSCI World (ACWI) | 0.6312 | +10.12% | 0.198 | 36.28% |
| ftse_china_a50 | FTSE China A50 ETF (2822.HK) | 0.3562 | +53.91% | 0.008 | 26.72% |
| eurostoxx_50 | EURO STOXX 50 (STOXX50E) | 0.2804 | +15.44% | 0.120 | 15.93% |

(+53.91%) and significant ($p = 0.008$). In other markets, while alpha remains positive, the higher $p$-values suggest weaker evidence of systematic outperformance.

Beta values in the range of 0.28 to 0.67 indicate moderate market exposure. This shows that the strategy is not entirely market-neutral, but it is also not simply tracking index movements. This aligns with our use of short-term technical forecasts rather than macro-driven positions.

The R-squared values, which range from 15.93% to 44.04%, show that a significant portion of return variation is not explained by CAPM, particularly in the European and Chinese markets. This points to a meaningful degree of idiosyncratic return generation, consistent with a model that is extracting useful trading signals beyond traditional market risk.

## H.2  THE FAMA–FRENCH 6-FACTOR MODEL

We further evaluated performance using the Fama-French 6-factor model. This model expands on CAPM by adding five more factors: market, size, value, profitability, investment, and momentum:

$$R_p - R_f = \alpha + \beta_1 (R_m - R_f) + \beta_2 \cdot \text{SMB} + \beta_3 \cdot \text{HML} + \beta_4 \cdot \text{RMW} + \beta_5 \cdot \text{CMA} + \beta_6 \cdot \text{WML} + \epsilon$$

Table 11: Fama–French 6-Factor Regression Results.

| Dataset | Annualized Alpha ($\alpha$) | Alpha p-value | R-squared |
|---|---|---|---|
| dowjones_30 | +0.18% | 0.985 | 38.77% |
| stocknet | +9.57% | 0.213 | 39.37% |
| ftse_china_a50 | +73.40% | 0.001 | 13.81% |
| eurostoxx_50 | +13.67% | 0.184 | 12.35% |

The Fama-French 6-factor analysis gives a more granular view of performance. While positive alpha persists in all datasets, their signficance varies, ranging from 0.001 in China A50 (very significant) to 0.985 (not significant). One possible reason for the lack of significance could be due to the short period horizon of our test dataset (1 year).

The R-squared values, ranging from 12.35% to 39.37%, indicate that even after accounting for a broader set of risk factors beyond CAPM, a significant share of return variation remains unexplained, which is good. In our work, technical analysis is typically designed for short-term optimization, capturing brief momentum, reversal, or volume-based effects. Because of this short-term focus, our model is not expected to align closely with long-horizon economic models like CAPM or Fama-French, which are typically evaluated over months or quarters.

Overall, the results demonstrate that our approach does offer some complementary value to CAPM and Fama-French 6 factors to be used as an interpretable, forward-looking portfolio signal.

