# OpenReview forum: "Reasoning on Time-Series for Financial Technical Analysis"
_ICLR.cc/2026/Conference — ICLR 2026 Poster_

### Official Review · Reviewer_6br1 · 2025-10-23

**Soundness:** 2
**Presentation:** 2
**Contribution:** 2
**Rating:** 4
**Confidence:** 4

**Summary:**

This paper presents verbal technical analysis (VTA) that integrates verbal reasoning through LLMs with latent time series forecasting to achieve interpretable financial prediction. The authors argue that existing LLM-based approaches primarily analyze textual financial information (e.g., reports or news), and VTA is distinct from them since it focuses on time series reasoning, transforming numerical data into natural language descriptions. Through extensive experiments across multiple stock datasets, VTA demonstrates state-of-the-art forecasting accuracy. In addition, VTA achieves superior expert evaluation scores compared to GPT and DeepSeek.

**Strengths:**

- Each component of the framework (time series reasoning, forecasting, etc) is well-motivated and properly designed.
- Experiments are comprehensive, covering a wide range of datasets, baselines, and metrics.
- The expert-based reasoning evaluation is practically valuable.
- The authors provided their code anonymously for reproduciblity.

**Weaknesses:**

- Some parts of the paper are unclear:
  - The discussion of Figure 1 should be expanded. Currently, the y-axis and evaluation criteria are ambiguous. It would be helpful to explicitly describe what each score measures and how they were computed.
  - The notation *s* in Eq. (10) and Figure 3 should be aligned for consistency.
  - Some terms in Table 5 (e.g., volatility, drawdown) should be clearly defined in the main text for non-finance readers.
- The scope of the paper needs justification:
  - The paper focuses only on financial datasets. Please elaborate on why the financial domain is particularly challenging (if so) for time series reasoning.
  - Please discuss whether VTA could generalize to other domains (e.g., healthcare, energy).
- There already exists a recent work that explores LLM-based time series explanation. The paper should clearly position VTA with respect to this approach by discussing or empirically comparing it.
  - TimeCAP: Learning to Contextualize, Augment, and Predict Time Series Events with Large Language Model Agents (AAAI 2025)


In summary, while the idea is interesting and timely, the paper needs a clearer positioning with the existing work. ALso, several aspects of the presentation could be improved.

**Questions:**

- How does VTA perform when applied to non-financial time series data?
- Does VTA show consistent reasoning patterns across markets (U.S. vs. China vs. Europe)?

In addition, please refer to the weaknesses above.

---

> ### Author Response · Authors · 2025-11-21
>
> Dear reviewer 6br1, thank you for your time in reviewing our work. We have addressed all your points below, and updated the paper according to your comments. If there are more concerns or something was not explained well, please continue to engage us till the end of the review period. If you think that our idea is interesting and timely, please also consider raising our score, as we will truly appreciate it.
>
> --
>
> **W1:** _Some parts of the paper are unclear._
>
> - We shifted Figure 1 to the main text, under **Reasoning Quality** in the Results section. The evaluation criteria can be found in **Appendix C.1**.
>
> - We have updated the notation in Eq. (10) to align it with Figure 3 (Figure 2 in updated version).
>
> - The definition of the terms in Table 5 (_e.g.,_ volatility, drawdown) are added in **Appendix H**, due to limited space in the main text.
>
> --
>
> **W2:** _Please discuss whether VTA could generalize to other domains._
>
> **W3:** _There already exists a recent work that explores LLM-based time series explanation…_
>
> These two weaknesses can actually be explained together for a better understanding of our work.
>
> The paper focuses on financial datasets as it contains known, interpretable signals which are backed by literature on Technical Analysis (Line 38). For example, stock prices tend to **move on momentum**, **move towards its moving average**, **have support and resistance levels**, _etc._ In other domains, the time-series do not contain such interpretable signals, hence it is not as meaningful to reason over them. For these domains, we show some examples in response to your question below.
>
> The TimeCAP work tackles time-series reasoning using a different approach. To forecast a specific time-series (_e.g.,_ rainfall), TimeCAP “contextualizes” the data by summarizing external information (_e.g.,_ temperature, humidity, etc.) using an LLM. During test-time, it retrieves the summary with the most similar conditions to “augment” the data with in-context examples. The final “prediction” produce reasoning on the time-series data by referencing these in-context examples. This is similar to a Retrieval-Augmented-Generation (RAG) work, but in the time-series domain.
>
> To draw an analogy to language tasks research, VTA is akin to a work on reasoning LLMs whereas TimeCAP is akin to a work on RAG. We believe both are crucial lines of work that push forward the frontier in time-series reasoning research. However, it might not be fair to compare their results directly as:
> 1. TimeCAP uses more information as inputs compared to our VTA model.
> 2. TimeCAP forecasts classification labels, while VTA forecasts the full time-series trajectory.
>
> **We have also now updated the introduction of our work to position ourselves against TimeCAP.**

---

> ### Author Response · Authors · 2025-11-21
>
> **Q1:** _How does VTA perform when applied to non-financial time series data?_
>
> As explained earlier, it is not as meaningful to reason over other time-series data, as they do not contain rich interpretable signals. To demonstrate this, we ran VTA over the healthcare (ILI) and energy (ETTh1) datasets, and show some examples of the reasoning traces:
>
> **Healthcare:**
>
> _The data shows a clear pattern of fluctuation, with a general trend of decreasing % Weighted ILI values over the last few weeks. The most recent values are in the negative range, indicating a decline in influenza-like illness. Given this trend, it's reasonable to predict a continuation of this decline, but with some variability. The last few weeks have shown a steady decline, so we can expect this trend to continue, but with some fluctuation. I will predict a gradual decrease with some variability._
>
> **Energy:**
>
> _The given data is the standardized transformer measurements (OT) for 96 hours. The task is to predict the oil temperature for the next 24 hours. Given the nature of the data and the task, a simple moving average or a trend extrapolation can be used for prediction. However, for this example, I will use a basic approach of taking the last value of the OT series as a reference and slightly adjusting it based on the recent trend. The last value of the OT series is -0.954. The recent trend shows a slight decrease in the last few values. I will predict a slight decrease in the next 24 hours, considering the recent trend and the range of the data._
>
> **Finance (Ours):**
>
> _The stock prices have been increasing over the past 10 days, with a recent peak on day 9. The moving averages suggest a slight upward trend, but the momentum and Williams %R indicate some overbought conditions. The Bollinger Bands show the current price is above the middle band, but not yet at the upper band, suggesting a potential for a pullback. However, the RSI and MACD are still in positive territory, indicating a bullish trend. Given the recent increase and the positive indicators, the stock is likely to continue its upward trend, but with some potential for a pullback. Predicting the exact prices is challenging, but we can use the recent trend and indicators to make an educated guess. The prices are likely to remain above the recent lows but not necessarily break new highs. We can assume a slight increase from the last price, with some potential for a pullback._
>
> As we can see, there is not much intrinsic information for the model to reason over in the other domains, other than performing simple trend extrapolation. For the financial domain, note that the indicators (RSI, MACD, _etc._) are all calculated directly from the input financial time-series and are not external data.
>
> --
>
> Furthermore, we can look at the performance comparison:
> |  | ILI |  | ETTh1 |  |
> |---|:---:|:---:|:---:|:---:|
> |  | MSE | MAE | MSE | MAE |
> | Informer | 1.7372 | 0.8669 | 0.3128 | 0.4848 |
> | Transformer | 1.6827 | 0.8122 | 0.2505 | 0.4360 |
> | Crossformer | 1.3320 | 0.7270 | 0.1298 | 0.2914 |
> | TSMixer | 2.1163 | 0.9504 | 0.0877 | 0.2375 |
> | Reformer | 1.5554 | 0.8134 | 0.2422 | 0.4020 |
> | LightTS | 2.3121 | 1.0612 | 0.0625 | 0.1921 |
> | DLinear | 2.9160 | 1.3933 | 0.0404 | 0.1499 |
> | FiLM | 2.6761 | 1.3856 | 0.0423 | 0.1543 |
> | Non-stationary | 1.4328 | 0.8538 | 0.0403 | 0.1530 |
> | MICN | 1.3431 | 0.8732 | 0.3063 | 0.4649 |
> | Autoformer | 1.7554 | 1.0350 | 0.0736 | 0.2146 |
> | TimesNet | **1.0795** | **0.6910** | 0.0410 | 0.1572 |
> | CALF | 1.4442 | 0.7409 | 0.0361 | 0.1411 |
> | VTA (Ours) | 1.4366 | 0.7090 | **0.0346** | **0.1374** |
>
> We first observe that the healthcare (ILI) and energy (ETTh1) datasets do not follow the same performance patterns we saw in financial time-series, which could be attributed to having different defining characteristics. For example, models that benefit from trend-seasonal decomposition on financial data do not show the same advantage here, likely because ILI and ETTh1 do not exhibit the same cyclical structure.
>
> In these domains, especially on the ILI dataset, our VTA model does not exhibit performance that is too far away from the time-series LLM model (CALF). The additional reasoning traces provide limited benefit when there is not much complex signals to reason over outside of simple trend extrapolation.

---

> > ### Author Response · Authors · 2025-11-21
> >
> > **Q2:** _Does VTA show consistent reasoning patterns across markets (U.S. vs. China vs. Europe)?_
> >
> > Yes. VTA show consistent reasoning patterns across all markets, as they are trained in the same manner. Here are some examples:
> >
> > **China A50:**
> >
> > _The stock prices have shown a range from a low of 0.7938 to a high of 1.099, with a mean price of 0.9407. The moving averages suggest a slight upward trend, with the simple moving average and exponential moving average converging around 0.9923. The momentum indicator has been positive but decreasing, and the RSI has been oscillating, indicating a mix of bullish and bearish signals. The MACD line is positive but small, suggesting a slight bullish trend. The Williams %R and CCI show that the stock is not overbought or oversold, and the Bollinger Bands show the stock price is within the upper and lower bands, indicating no extreme volatility. The stochastic indicators are also in a neutral range. Given these factors, the stock is likely to continue its upward trend but with some volatility. Predicting the next 10 days, I would expect the stock to continue to rise but with some fluctuations. I'll assume a slight increase in the upward trend, but not too aggressive, to account for the neutral to slightly bullish signals._
> >
> > **EUROSTOXX 50:**
> >
> > _Given the data, we can observe that the stock price has been generally increasing over the past 10 days, with the highest price on day 8. The moving averages suggest a slight upward trend, and the RSI and Williams %R indicate that the stock is not overbought or oversold, but rather in a healthy range. The MACD line is positive, indicating a bullish trend. The Bollinger Bands show that the stock price is within the upper band, suggesting that the stock might be due for a correction. However, the stochastic indicators are not at extreme levels, indicating that a correction might not be imminent. Considering the recent trend and the indicators, it's reasonable to predict a slight increase in the next 10 days, but not a large one. We can use a combination of the recent trend and the Bollinger Bands to make our prediction._

---

> > > ### Comment · Reviewer_6br1 · 2025-11-23
> > >
> > > Thank you for addressing my concerns. I look forward to seeing the additional results, case studies, and discussion of relevant work in the final version of the paper. I have updated my score accordingly.

---

> > > > ### Author Response · Authors · 2025-11-28
> > > >
> > > > Dear reviewer 6br1, thank you sincerely for your supportive comments and for increasing our score.
> > > >
> > > > The additional results, case studies and discussion of relevant work will be included in the final version of the paper.

---

### Official Review · Reviewer_DJah · 2025-10-30

**Soundness:** 2
**Presentation:** 3
**Contribution:** 2
**Rating:** 4
**Confidence:** 3

**Summary:**

This paper proposes Verbal Technical Analysis (VTA) — a hybrid reasoning framework that aims to make time-series stock forecasting interpretable by combining:
- A verbal reasoning LLM fine-tuned via a proposed Time-GRPO objective (using inverse-MSE reward);
- A latent time-series backbone model conditioned on the reasoning outputs through joint conditional training.

Experiments across U.S., Chinese, and European stock datasets suggest VTA achieves better MSE/MAE than prior time-series LLMs (e.g., Time-LLM, CALF) and generates reasoning traces preferred by human financial experts. The paper also evaluates downstream portfolio performance using Sharpe ratio and CAPM/Fama-French models.

**Strengths:**

pros:
1. Interesting conceptual direction — bridging LLM reasoning and numerical forecasting is a timely and under-explored research frontier, particularly for financial data.

2. Clear problem motivation — the authors highlight the lack of interpretable time-series reasoning methods and position VTA as a step toward explainable financial LLMs.

3. Engineering completeness — the framework, ablations, and appendices (notably the extensive tables on pages 6–9 and expert evaluation in Appendix C) are well-executed and professional.

4. Human evaluation effort — 25 finance professionals were surveyed; results show statistically significant preference for VTA reasoning over GPT-4.1 and DeepSeek-R1 (Table 7).

5. Empirical results — consistent improvement across four datasets; the method performs competitively even under portfolio-level evaluation (Table 5).

**Weaknesses:**

1. The “Time-GRPO” formulation largely reuses GRPO (Shao et al., 2024) with an inverse-MSE reward; conceptually it is a direct extension rather than a fundamentally new RL objective. The “joint conditional training” (Figure 3) borrows heavily from classifier-free guidance (Ho & Salimans 2022) and conditional diffusion principles. The novelty lies more in application assembly than in algorithmic innovation.
2. The paper never establishes whether the verbal reasoning causally improves forecasting accuracy, or merely adds correlated signals. The conditioning between textual reasoning and time-series output feels heuristic; no ablation demonstrates whether removing reasoning entirely (vs. random text conditioning) degrades performance beyond noise.
3. The “expert evaluation” metrics (Clarity, Depth, etc.) are subjective and lack reproducibility; no public dataset of reasoning traces is released. The forecasting datasets are relatively small (StockNet, A50, EUROSTOXX 50). Improvements of ≈2–3% MSE over CALF (Table 2) may not be statistically meaningful beyond the reported seeds.
4. Does not compare to simpler text-free multimodal adapters (e.g., cross-attention-based transformers with interpretable heads). The authors claim SOTA forecasting “and interpretability,” but do not quantify interpretability beyond expert survey averages.

**Questions:**

1. How sensitive is VTA to the choice of base LLM (e.g., Qwen vs Llama)?

2. Can the verbal reasoning traces be used independently for explainable trading, or are they only auxiliary features for the time-series decoder?

3. Would random or adversarial reasoning degrade forecasting performance — i.e., is the reasoning genuinely informative?

4. Is the “inverse-MSE reward” stable during optimization? Any reward hacking observed?

---

> ### Author Response · Authors · 2025-11-21
>
> Hi reviewer DJah, thank you for your detailed comments. We have addressed your concerns below as best as we could. The aim of our work was to unlock a new, underexplored capability of LLMs in time-series reasoning. Some points from the questions could also be found in the work. From this angle, we hope that you can consider raising our score based on our proposed contribution to the field. If there are more concerns, please continue to engage with us, and we are happy to address any further issues on our work.
>
> --
>
> **W1:** _…The novelty lies more in application assembly than in algorithmic innovation._
>
> We agree that our work builds on top of GRPO and conditional diffusion principles. This is not that different to how many deep-learning advances are built on top of other techniques, but demonstrating new capabilities. For example, LLMs themselves were built on top of existing transformer architectures, but they showed that by training transformers on next-word prediction over a large dataset, they were able to produce highly generalizable AI that can converse with us. Our contribution is not in proposing  new mathematical objectives or principles, but in showing how to adapt LLMs to a domain where it was shown to be weak at: verbal reasoning directly over time-series inputs.
>
> Prior recent works (Merrill et al., 2024; Chow et al., 2024 – cited in our work) have reported that LLMs are poor at time-series reasoning. Our work is one of the earliest to show that GRPO-style optimization, combined with domain-grounded rewards, can yield high-quality time-series reasoning. For example, in Figure 4 (Figure 3 in the updated version), we showed that the model can “learn” to reason over time-series as the training reward increases over time. This observation could potentially be useful for researchers to better understand the learning capabilities of LLMs and unlock new applications in the future. In this sense, our novelty lies in enabling a new capability (time-series reasoning), which is parallel to works that build the foundation for LLMs to do reasoning over images, reasoning over math problems, _etc._
>
> --
>
> **W2:** _… no ablation demonstrates whether removing reasoning entirely (vs. random text conditioning) degrades performance beyond noise._
>
> Thank you for this suggestion. We have now included some reasoning ablations in the response to your question below.
>
> --
>
> **W3:** _The “expert evaluation” metrics (Clarity, Depth, etc.) are subjective and lack reproducibility; no public dataset of reasoning traces is released … Improvements of ≈2–3% MSE over CALF (Table 2) may not be statistically meaningful beyond the reported seeds._
>
> We have previously used LLM-as-a-judge for evaluation but this was not reported due to recent criticism of LLM as evaluators by many reviewers. However, we now include that in our updated paper (Appendix G), so that we have an evaluation method that is less subjective and reproducible.
>
> We did not save every single reasoning trace due to memory concerns. However, in general, most reasoning traces look similar to what was shown in the case studies. Our model was trained to effectively use the technical indicators, whereas the LLMs we compared with are not trained to do so.
>
> There was a statistical significance test previously reported in Appendix D. This was done over 10 runs across different seeds, and our performance is found to be statistically significant. The improvement over CALF was not huge, but consistent. We also note that the focus of this work was to produce the reasoning traces rather than optimizing for the best forecasts, hence we did not experiment with forecast settings such as bigger parameter size, _etc._
>
> --
>
> **W4:** _Does not compare to simpler text-free multimodal adapters (_e.g.,_ cross-attention-based transformers with interpretable heads). The authors claim SOTA forecasting “and interpretability,” but do not quantify interpretability beyond expert survey averages._
>
> We did compare with cross-attention-based transformers such as Crossformer, Autoformer, _etc._ in our results. There could be some difference between our understanding of interpretability: Whereas the attention-based models could structural interpretability via attention weights, these mechanisms show feature importance but not verbal explanations. In contrast, our VTA framework introduces semantic interpretability: it generates explicit natural-language reasoning traces that describe the rationale behind each forecast, optimized directly for predictive correctness. As such, the best way we can evaluate these reasoning traces is through expert survey or LLM-as-a-judge (which we have now included based on your earlier comment). This is also what is generally done in LLM reasoning literature.

---

> ### Author Response · Authors · 2025-11-21
>
> **Q1:** _How sensitive is VTA to the choice of base LLM (e.g., Qwen vs Llama)?_
>
> The ablation study on base LLM was reported in Table 3. Among the 3 LLMs we tried, Qwen-2.5-7B is better than Llama-3.1-8B, which is better than Qwen-2.5-3B. While the 3B Qwen model has significantly worse performance before fine-tuning, the model performance for all base LLMs becomes quite close after going through the fine-tuning process for VTA.
>
> --
>
> **Q2:** _Can the verbal reasoning traces be used independently for explainable trading, or are they only auxiliary features for the time-series decoder?_
>
> Yes, the verbal reasoning traces can be used independently by directly asking the LLM to produce the time-series outputs. Their performance was also reported in the ablation study in Table 3. The performance without the time-series component is found under “RL for reasoning”, before the conditioning step.
>
> --
>
> **Q3:** _Would random or adversarial reasoning degrade forecasting performance — i.e., is the reasoning genuinely informative?_
>
> To study the contribution of the reasoning component on forecasting performance, we added some additional experiments by corrupting the reasoning trace to observe their impact. This was done in two ways: **(1) Adversarial:** We add adversarial noise to the technical indicator values (noise is added in multiples of standard deviations); **(2) Remove Information:** Within the prompt, we force the LLM to not utilize certain indicators in its reasoning.
>
> | **Adversarial Noise** | **MSE** |  | **Indicators Removed** | **MSE** |
> |:---:|:---:|---|:---:|:---:|
> | **0** | **0.0659** |  | **0** | **0.0659** |
> | 1 | 0.0668 |  | 1 | 0.0660 |
> | 2 | 0.0665 |  | 2 | 0.0664 |
> | 3 | 0.0674 |  | 3 | 0.0671 |
> | 4 | 0.0673 |  | 4 | 0.0675 |
> | 5 | 0.0672 |  | 5 | 0.0679 |
>
> We find that removing the indicators leads to a clear degradation in performance, demonstrating that having the additional indicator signals from the reasoning traces provide genuinely useful guidance to the model. Adding adversarial noise also reduces overall performance but does not yield a consistent trend. A possible explanation is that, during joint conditional training, the model gradually learns to rely more heavily on the time-series forecasting component once it detects that the reasoning signals have become unreliable.
>
> This has been a very helpful suggestion to our work. **We have now included it as an additional paragraph in the updated paper.**
>
> --
>
> **Q4:** _Is the “inverse-MSE reward” stable during optimization? Any reward hacking observed?_
>
> The reward was stable during optimization, and no reward hacking was observed.
>
> The reward curve was reported in Figure 4 (Figure 3 in the updated version). Typically, reward hacking would produce these patterns in the curve:
>
> 1. Sudden, vertical spikes when a “hack” is found.
> 2. Reward increases indefinitely, as the model is exploiting some artifact in the reward definition.
> 3. Variance spirals outward, as the reward gets noisier and noisier.
>
> From the figure, we see that none of these are happening. Instead, we see a gradual upward trend at the start, which shows that the model is learning how to reason over time-series. After about 250 steps, the curve plateaus, which shows stable convergence. There are some oscillations around the plateau – these can be attributed to the “exploration” process of the reinforcement learning, and also because GRPO is trained in groups, resulting in some natural variation. The KL constraint (Equation 3) also helps to push the model back whenever it deviates too far from the learnt policy.

---

> ### Author Response · Authors · 2025-11-27
>
> Dear reviewer DJah, to address your point about having no public dataset of reasoning traces, we have run our VTA model over the StockNet dataset to reproduce the reasoning traces. They can be found here, together with the forecasts:
>
> https://anonymous.4open.science/r/vta-supplementary-4D74
>
> --
>
> Additionally, we would like to further address your point on interpretability. Our VTA framework mainly addresses natural-language interpretability. This is in contrast to other techniques like interpretable attention weights or SHAP values, which can highlight feature importance. VTA **does not** rank features on their importance, but it explains how the features combine to produce the forecasts. To highlight their differences clearly, we performed some experiments:
>
> Following your comment, we ran a simple cross-attention transformer and obtained the attention weights. In addition, we also use KernelSHAP to produce SHAP values:
>
> | **Feature**                        | **Attention Weights** |   | **Feature**                      | **SHAP Values** |
> |------------------------------------|-----------------------|---|----------------------------------|-----------------|
> | Simple Moving Average   (SMA)      | 0.2134                |   | Simple Moving Average (SMA)      | 0.00975         |
> | Exponential Moving Average   (EMA) | 0.2068                |   | Exponential Moving Average (EMA) | 0.00853         |
> | Average Directional Index   (ADX)  | 0.1043                |   | Stochastic Oscillator            | 0.00727         |
> | MACD Line                          | 0.0903                |   | Commodity Channel Index (CCI)    | 0.00675         |
> | Bollinger Bands                    | 0.0777                |   | Williams %R                      | 0.00650         |
> | Commodity Channel Index   (CCI)    | 0.0664                |   | Bollinger Bands                  | 0.00625         |
> | Relative Strength Index   (RSI)    | 0.0624                |   | MACD Line                        | 0.00571         |
> | Williams %R                        | 0.0607                |   | Average Directional Index (ADX)  | 0.00558         |
> | Stochastic Oscillator              | 0.0607                |   | Relative Strength Index (RSI)    | 0.00514         |
> | Momentum                           | 0.0573                |   | Momentum                         | 0.00500         |
>
> We see here that both techniques are in agreement on the top 2 important features (_i.e.,_ SMA, EMA). Note that the attention weights show where the model is focusing on when making its forecasts, but is not considered as interpretability in some literature`[1]`.
>
> To evaluate this, we re-run the model repeatedly, removing features one-by-one from most to least important, to see the impact on test MSE (_i.e.,_ _faithfulness_ metric `[2]`):
>
> | **Feature Removed**                | **MSE** | **MSE Change** |
> |------------------------------------|:-------:|:--------------:|
> | None                               |  1.401  |        -       |
> | Simple Moving Average   (SMA)      |  2.089  |     -49.2%     |
> | Exponential Moving Average   (EMA) |  6.213  |     -197.4%    |
> | Average Directional Index   (ADX)  |  6.140  |      1.2%      |
> | MACD Line                          |  6.207  |      -1.1%     |
> | Bollinger Bands                    |  6.475  |      -4.3%     |
> | Commodity Channel Index   (CCI)    |  6.610  |      -2.1%     |
> | Relative Strength Index   (RSI)    |  6.456  |      2.3%      |
> | Williams %R                        |  6.388  |      1.1%      |
> | Stochastic Oscillator              |  6.849  |      -7.2%     |
> | Momentum                           |  7.348  |      -7.3%     |
>
> Here, we see that even though SMA was the top ranked feature, removing EMA result in an even bigger performance drop. _(Continued below)_

---

> ### Author Response · Authors · 2025-11-27
>
> However, we suspect that the usefulness of each feature is not independent. To verify this, we repeat the experiments a few more times, but we drop the features randomly:
>
> | **Feature Removed**                | **MSE** | **MSE Change** |   | **Feature Removed**              | **MSE** | **MSE Change** |
> |------------------------------------|:-------:|:--------------:|---|----------------------------------|:-------:|:--------------:|
> | None                               |  1.401  |        -       |   | None                             |  1.401  |        -       |
> | Williams %R                        |  1.568  |     -12.0%     |   | Momentum                         |  1.558  |     -11.2%     |
> | Bollinger Bands                    |  1.379  |      12.1%     |   | MACD Line                        |  1.448  |      7.1%      |
> | Commodity Channel Index   (CCI)    |  1.235  |      10.4%     |   | Average Directional Index (ADX)  |  1.506  |      -4.0%     |
> | Simple Moving Average   (SMA)      |  1.674  |     -35.5%     |   | Bollinger Bands                  |  1.309  |      13.0%     |
> | MACD Line                          |  1.515  |      9.5%      |   | Exponential Moving Average (EMA) |  1.387  |      -5.9%     |
> | Stochastic Oscillator              |  1.703  |     -12.4%     |   | Commodity Channel Index (CCI)    |  1.308  |      5.7%      |
> | Relative Strength Index   (RSI)    |  1.402  |      17.7%     |   | Williams %R                      |  1.242  |      5.0%      |
> | Exponential Moving Average   (EMA) |  6.841  |     -388.0%    |   | Simple Moving Average (SMA)      |  6.582  |     -430.0%    |
> | Momentum                           |  7.626  |     -11.5%     |   | Relative Strength Index (RSI)    |  6.821  |      -3.6%     |
> | Average Directional Index   (ADX)  |  7.343  |      3.7%      |   | Stochastic Oscillator            |  7.346  |      -7.7%     |
>
> The experiment on the right highlights an important point: The model suffers the most performance drop after **both** SMA and EMA are removed. In Finance, this is because the interaction between these 2 indicators is a useful signal. Standard interpretation methods like attention weights and SHAP usually only show individual feature importance, but do not capture the interactions between features.
>
> --
>
> We now look at an example of VTA’s verbal interpretability:
>
> **Explanation:** _The stock price has shown a range from a low of 1.56 to a high of 1.736, with a mean of 1.637. The moving averages suggest a slight_ **upward trend**_, with the simple moving average (SMA) and exponential moving average (EMA) both indicating a recent increase. The momentum indicator shows a positive trend, while the RSI indicates a slight_ **overbought** _condition. The MACD line shows a slight decrease, which could indicate a_ **potential reversal**_. The Williams %R and CCI suggest the stock may be_ **overbought**_, while the Bollinger Bands show the stock price is within the upper band, indicating a_ **possible correction**_. The Stochastic Oscillator also suggests the stock may be_ **overbought**_._
>
> **Ground Truth:** _[0.0664, 0.0374, 0.0229, 0.0247, 0.0125, 0.0161, 0.0259, 0.0186, 0.0124, 0.0064]_ _(Downward trend)_
>
> Here, we see that VTA does not rank features, but consider all of them as a whole. Interestingly, while the two “most important” features SMA and EMA suggest an upward trend, most other indicators show a downward pressure (overbought = will go down). Looking at the ground truth series, the time-series did indeed go down. Simple methods like attention weights might not be able to capture this type of interpretability.
>
> From this, we point out that the verbal interpretability of VTA and interpretability based on attention weights are not directly comparable. Typically, verbal reasoning traces are evaluated using expert survey averages `[3]` or LLM-as-a-judge, both of which we have reported in our work.
>
> --
>
> `[1]` Jain, Sarthak, and Byron C. Wallace. "Attention is not explanation." arXiv preprint arXiv:1902.10186 (2019).
>
> `[2]` Rezaei, Shahbaz, and Xin Liu. "Explanation space: A new perspective into time series interpretability." arXiv preprint arXiv:2409.01354 (2024).
>
> `[3]` Li, Shuqi, et al. "PEN: prediction-explanation network to forecast stock price movement with better explainability." Proceedings of the AAAI Conference on Artificial Intelligence. Vol. 37. No. 4. 2023.
>
> --
>
> To verify our experiments, we also release the additional codes here: https://anonymous.4open.science/r/vta-supplementary-4D74
>
> From these experiments, we have answered your point on comparing with cross-attention-based transformers with interpretable heads. It could be possible that we did not understand your point fully. If this is so, please continue to engage us and we are happy to answer any further questions. Thank you so much!

---

### Official Review · Reviewer_ndFL · 2025-10-31

**Soundness:** 3
**Presentation:** 3
**Contribution:** 3
**Rating:** 4
**Confidence:** 3

**Summary:**

The authors propose a method for training a model that can reason over financial time series. The model works by jointly learning over textual and numerical representations of time series data. The authors conduct an evaluation by comparing the method to several other LM-based time series forecasting methods and conduct a human evaluation to measure the plausibility of the model's generated traces.

**Strengths:**

- Well-written, easy to understand
- This paper contrasts with others in the area by combining neural representations of LMs with text embeddings. Most other papers focus only on time series expressed as text or on neural embeddings, not both.
- Strong choice of baselines and third-party datasets
- The use of experts for annotation is useful and lends confidence towards the model's performance

**Weaknesses:**

On the whole I think this is a very strong paper. However, without details on how baselines were tuned it's difficult for me to have confidence in the method's performance. If the authors can provide this information then I'll happily increase my score to an accept.

To be clear, my primary concern is that the proposed method could be outperformed by (1) simple statistical methods or (2) strongly hyperparameter-tuned baselines.

**Questions:**

- Why do we need dedicated financial time series reasoning? Why aren't the other times series LLMs useful for this job?
- How does performance compare to simpler models? "Are Language Models Actually Useful for Time Series Forecasting?" Tan et al. 2024 shows that many LLM-based methods for time series forecasting are outperformed by simple baselines (like mean prediction).
- Could the authors provide more information on how they tuned their baselines? Im concerned that the baseline models could outperform the proposed method given sufficient tuning.

---

> ### Author Response · Authors · 2025-11-21
>
> Hi reviewer ndFL, thank you for your kind review. We have addressed your questions below. Feel free to engage us if any points are not clear, and we are happy to continue the discussion with you. Please also let us know if there is anything more we can do for you to increase our score further.
>
> --
>
> **Q1:** _Why do we need dedicated financial time series reasoning? Why aren't the other times series LLMs useful for this job?_
>
> This point was addressed in the work. Current time-series LLMs like TimeLLM, CALF, typically modify the embedding space of the LLMs, making them lose their natural language reasoning capabilities. Also, financial time-series consist of known, interpretable signals that many financial practitioners study, which do not exist in other time-series. For example, stock prices tend to **move on momentum**, **move towards its moving average**, **have support and resistance levels**, _etc._ Other time-series typically do not have such intrinsic signals, and general time-series models mostly do trend extrapolation or pattern matching on past data.
>
> --
>
> **Q2:** _How does performance compare to simpler models?_
>
> The performance comparison with simpler models is as follows:
> |  | MSE | MAE |
> |---|---|---|
> | MeanP | 0.0938 | 0.2127 |
> | ARIMA | 1.2614 | 3.0389 |
> | AutoARIMA | 0.0847 | 0.1949 |
> | VTA (Ours) | **0.0659** | **0.1701** |
>
> Our VTA model was able to beat the simpler models such as mean prediction, ARIMA, or AutoARIMA.
>
> This could be linked to our answer to your previous question: financial time-series follow more complicated rules (_e.g.,_ they might move on momentum, or move within dynamic ranges), in which simple models might not be able to capture. This might be in contrast with other time-series like electricity usage, which generally follow more simple trends (_e.g.,_ high in the day, low at night), and complicated models might overfit to such datasets as opposed to simple models.
>
> --
>
> **Q3:** _Could the authors provide more information on how they tuned their baselines?_
>
> The hyperparameters were tuned over a limited search space using a script file, which runs the model multiple times under different settings.
>
> Since each model contains numerous hyperparameters, including (but not limited to): learning rate, dropout, attention heads, encoder layers, decoder layers, hidden sizes, it is not feasible to strongly tune the models across all possible parameter combinations. For the hyperparameters that affect the model architecture, such as number of attention heads, encoder/decoder layers, we mostly chose the values that was used in the original paper. For the more sensitive parameters such as learning rate, hidden sizes, dropout, we selected a few configurations and run the model on them. We chose the best configuration by selecting the run with the highest validation MSE.
>
> Overall, we find that the MSE performance is largely consistent across different hyperparameters, only fluctuating within a small range (_e.g._ 2-3% difference). One important note is that from the results table (Table 2), we see a consistent relative trend across all datasets, where the stronger model on one dataset tends to perform as well on other datasets. This also shows that the performance trend is robust and not largely affected by hyperparameter sensitivity (_i.e.,_ if any one baseline was under-tuned on a specific dataset, we would expect its relative performance to fluctuate across the different datasets).

---

> > ### Comment · Reviewer_ndFL · 2025-11-24
> >
> > On Q3 can you just confirm that all methods tested (including the one you propose) had the same process for optimizing hyperparameters?

---

> > > ### Author Response · Authors · 2025-11-25
> > >
> > > Yes, we confirm that.

---

> > > > ### Comment · Reviewer_ndFL · 2025-11-25
> > > >
> > > > Then sure, I'll bump my rating.

---

> > > > > ### Author Response · Authors · 2025-11-28
> > > > >
> > > > > Dear reviewer ndFL, thank you sincerely for your supportive comments and for increasing our score.

---

### Official Review · Reviewer_jfgi · 2025-11-01

**Soundness:** 3
**Presentation:** 3
**Contribution:** 2
**Rating:** 6
**Confidence:** 4

**Summary:**

This paper proposes a novel framework called Verbal Technical Analysis (VTA), which combines verbal (LLM-based) and latent (time-series model-based) reasoning for interpretable financial time-series forecasting. The framework introduces a reinforcement learning training pipeline (Time-GRPO) to optimize reasoning quality, and conditioning mechanisms to guide forecasts based on reasoning outputs. VTA is evaluated on multiple datasets across U.S., Chinese, and European markets, achieving state-of-the-art results in both forecasting accuracy and interpretability.

**Strengths:**

The paper proposes a novel framework, Verbal Technical Analysis (VTA), which effectively integrates large language models with time-series forecasting to produce both accurate and interpretable stock predictions.

It introduces an innovative reinforcement learning objective (Time-GRPO) using inverse MSE as a reward to improve the quality of verbal reasoning aligned with forecast accuracy.

The model generates natural language reasoning traces that are rated highly by financial domain experts, adding a valuable layer of explainability for real-world financial decision-making.

The evaluation is rigorous, including ablation studies, expert human assessment, and practical validation through portfolio optimization, demonstrating both technical soundness and practical utility.

**Weaknesses:**

The proposed training pipeline is multi-staged and complex, involving cold-start RL, supervised fine-tuning, and joint conditional training, which may limit reproducibility and increase implementation overhead.

The paper does not compare VTA’s reasoning or performance against human-crafted technical analysis rules or human analysts, which would be a natural baseline for such a task.

The paper does not compare VTA with financial time seires LLMs, e.g., Kronous, which are related to financial analysis.

**Questions:**

How well does VTA scale to longer prediction horizons or more volatile, high-frequency financial time-series?

Does the verbal reasoning always align with the actual forecast, or are there cases where they contradict, and how is that handled?

How does VTA perform under market regime shifts or structural breaks, such as during crises or unexpected economic events?

---

> ### Author Response · Authors · 2025-11-21
>
> Hi reviewer jfgi, thank you for the review. Responding to this review took us longer as there were many additional experiments to run. We have addressed all your points to the best of our ability below. Please let us know if you have more questions, and do consider raising our score if we have answered to your satisfaction.
>
> --
>
> **W1:** _The proposed training pipeline is multi-staged and complex, involving cold-start RL, supervised fine-tuning, and joint conditional training, which may limit reproducibility and increase implementation overhead._
>
> To ensure full reproducibility, we have included the code in the work.
>
> Generally, most LLMs are fine-tuned over a multi-stage training pipeline. For example, ChatGPT was trained using supervised fine-tuning using annotated samples, then using reinforcement learning (RL) to capture human preferences `[1]`. DeepSeek was trained over a four-stage pipeline, which includes the cold start phase, reasoning RL, rejection sampling and scenario-based RL `[2]`. Most cross-modality reasoning LLMs are also trained over multiple stages `[3, 4]`, one stage to understand the modality and the other to learn reasoning, etc. Similarly to all these works, each of our stages serve a useful purpose, which we highlighted in our work.
>
> --
>
> **W2:** _The paper does not compare VTA’s reasoning or performance against human-crafted technical analysis rules or human analysts, which would be a natural baseline for such a task._
>
> Technical analysis rules (_e.g.,_ RSI thresholds, moving-average crossovers, MACD rules) are typically designed for trading signals rather than continuous forecasting. These rules produce binary buy/sell decisions and do not generate time-series predictions, making them not directly comparable to VTA’s forecasting objective. The signals themselves are also quite subjective, making it difficult to do a fair comparison that would not be criticized (_e.g.,_ if there are conflicting up/down signals appearing at the same time).
>
> --
>
> **W3:** _The paper does not compare VTA with financial time seires LLMs, e.g., Kronous, which are related to financial analysis._
>
> Not all financial LLMs were able to handle our task of predicting time-series from time-series input. For example, FinGPT `[5]`, InvestLM `[6]`, takes in text reports and generate a sentiment for the company. This is the problem we are addressing in our work. However, we can compare with Kronos, which is a time-series based model:
>
> |  | MSE | MAE |
> |---|---|---|
> | Kronos | 0.1168 | 0.2355 |
> | VTA (Ours) | **0.0659** | **0.1701** |
>
> Here, our model was able to outperform Kronos. One thing we noted was that Kronos was trained over much longer lookback windows (40-480 timesteps) and very high frequency data. This might not be a fair comparison for them, as our task looks at daily forecasts. We do not have high-frequency data available for experiments.
>
> --
>
> References:
>
> `[1]` Long Ouyang, Jeffrey Wu, Xu Jiang, Diogo Almeida, Carroll Wainwright, Pamela Mishkin, Chong Zhang, Sandhini Agarwal, Katarina Slama, Alex Ray, et al. Training language models to follow instructions with human feedback. Advances in neural information processing systems, 35: 27730–27744, 2022.
>
> `[2]` Daya Guo, Dejian Yang, Haowei Zhang, Junxiao Song, Ruoyu Zhang, Runxin Xu, Qihao Zhu, Shirong Ma, Peiyi Wang, Xiao Bi, et al. Deepseek-r1: Incentivizing reasoning capability in llms via reinforcement learning. arXiv preprint arXiv:2501.12948, 2025.
>
> `[3]` Haoyu Lu, Wen Liu, Bo Zhang, Bingxuan Wang, Kai Dong, Bo Liu, Jingxiang Sun, Tongzheng Ren, Zhuoshu Li, Hao Yang, et al. Deepseek-vl: towards real-world vision-language understanding. arXiv preprint arXiv:2403.05525, 2024.
>
> `[4]` Winnie Chow, Lauren Gardiner, Haraldur T Hallgr´ımsson, Maxwell A Xu, and Shirley You Ren. Towards time series reasoning with llms. arXiv preprint arXiv:2409.11376, 2024.
>
> `[5]` Hongyang Yang, Xiao-Yang Liu, and Christina Dan Wang. Fingpt: Open-source financial large language models. arXiv preprint arXiv:2306.06031, 2023a.
>
> `[6]` Yi Yang, Yixuan Tang, and Kar Yan Tam. Investlm: A large language model for investment using financial domain instruction tuning. arXiv preprint arXiv:2309.13064, 2023b.

---

> > ### Author Response · Authors · 2025-11-21
> >
> > **Q1:** _How well does VTA scale to longer prediction horizons or more volatile, high-frequency financial time-series?_
> >
> > Similar to our previous response, we do not have high-frequency data to run the experiments. To the best of our ability, we can run experiments on different prediction horizons (5, 20 days compared to 10 days in the paper) to observe the results.
> >
> > |  | _t=5_ |  | _t=20_ |  |
> > |---|---|---|---|---|
> > |  | MSE | MAE | MSE | MAE |
> > | Informer | 2.0313 | 0.9238 | 2.3914 | 1.0714 |
> > | Transformer | 1.3257 | 0.6966 | 0.1459 | 0.2605 |
> > | Crossformer | 1.0316 | 0.5804 | 1.5260 | 0.7750 |
> > | TSMixer | 2.1760 | 0.9435 | 1.8602 | 1.0255 |
> > | Reformer | 0.7805 | 0.5538 | 0.1276 | 0.2394 |
> > | LightTS | 0.4072 | 0.4035 | 0.4072 | 0.4035 |
> > | DLinear | 0.1707 | 0.3265 | 0.2680 | 0.3731 |
> > | FiLM | 0.0424 | 0.1352 | 0.1752 | 0.2896 |
> > | Non-stationary | 0.0376 | 0.1268 | 0.1610 | 0.2723 |
> > | MICN | 0.0403 | 0.1371 | 0.1890 | 0.2991 |
> > | Autoformer | 0.0372 | 0.1271 | 0.1891 | 0.3064 |
> > | TimesNet | 0.0390 | 0.1287 | **0.1267** | **0.2384** |
> > | CALF | 0.0363 | 0.1243 | 0.1446 | 0.2567 |
> > | VTA (Ours) | **0.0355** | **0.1228** | 0.1358 | 0.2473 |
> >
> > Here, we note that VTA did not obtain the best performance (but still strong results) on the longer time horizon dataset (20 days). Technical analysis is typically performed on shorter time horizons, and the technical indicator signals seem to be less effective over the longer time horizon. For long-term stock forecasting, it could be useful to also analyze the fundamental information of the companies.
> >
> > --
> >
> > **Q2:** _Does the verbal reasoning always align with the actual forecast, or are there cases where they contradict, and how is that handled?_
> >
> > This was the purpose of the joint conditioning module, which is to align the verbal reasoning with the actual forecast.
> >
> > Throughout the duration of the work, we randomly selected samples and checked how closely the reasoning matches the forecast. We “handled” this issue throughout the project by continuously refining our alignment component (_e.g.,_ trying different prompts, conditioning methods). That being said, the alignment is not completely perfect.
> >
> > We also note that the joint conditioning module takes in information from both the verbal and forecasting component. For example, it might ignore some information from the verbal reasoning if it finds that it does not contribute to the overall forecasting performance.
> >
> > --
> >
> > **Q3:** _How does VTA perform under market regime shifts or structural breaks, such as during crises or unexpected economic events?_
> >
> > To see the performance of VTA during crisis, we test the model on a 2020 Dow Jones dataset using training data from year 2016-2019. The 2020 test period coincides with the COVID-19 market crash, which should be “unexpected” for the model which is trained on previous years.
> >
> > |  | Dow Jones 2020 |  |
> > |---|---|---|
> > |  | MSE | MAE |
> > | Informer | 2.3797 | 1.0130 |
> > | Transformer | 1.5693 | 0.7983 |
> > | Crossformer | 1.3052 | 0.7135 |
> > | TSMixer | 1.2902 | 0.7707 |
> > | Reformer | 1.2954 | 0.7908 |
> > | LightTS | 0.5924 | 0.5426 |
> > | DLinear | 0.3503 | 0.4244 |
> > | FiLM | 0.2597 | 0.3413 |
> > | Non-stationary | 0.2234 | 0.3181 |
> > | MICN | 0.2206 | 0.3153 |
> > | Autoformer | 0.2295 | 0.3234 |
> > | TimesNet | 0.2235 | 0.3144 |
> > | CALF | 0.2119 | 0.3026 |
> > | VTA (Ours) | **0.2118** | **0.3024** |
> >
> > VTA remains the best model across the 2020 dataset. VTA could be less affected by market regime shifts or structural breaks as it only considers the near-term historical price information. These events would have a bigger impact on fundamental or factor investors (_e.g.,_ A company with strong retail growth in the past could be affected by COVID-19 and cause its price to drop. However VTA does not consider this growth information in the first place).
> >
> > However, we do notice the performance gap has become smaller to the non-reasoning based time-series LLM (CALF). It is possible that the technical indicators have become less reliable when the market structural relationships are unstable or disrupted. This goes beyond the scope of our work, but it is a very interesting phenomenon that we would like to explore further in the future.

---

> > > ### Author Response · Authors · 2025-11-22
> > >
> > > Note that the goal of our work is not simply to produce the best-performing forecasts. For financial practitioners, we contribute the following:
> > > 1. A tool that produces explanations for financial time-series analysis. Human investors can still combine it with their own knowledge before making investment decisions.
> > > 2. Most current LLM works only do explanations over text reports like news or earnings reports. Our work offers to explain a new, **non-correlated** trading signal for investors.

---

> > > > ### Author Response · Authors · 2025-11-27
> > > >
> > > > Hi reviewer jfgi, we thank you again for your insightful comments.
> > > >
> > > > In addition to the multiple experiments we have done in response to your comments, we have also made the following changes to our work:
> > > >
> > > > 1. We modified the introduction to position our work better against current LLM time-series explanation works, thanks to comments from other reviewers.
> > > > 2. We studied the contribution of the reasoning component on forecasting performance by corrupting the reasoning traces and observing their impacts.
> > > > 3. We discussed the generalizability of VTA to other time-series domains such as healthcare and energy.
> > > > 4. We added an additional evaluation method using LLM-as-a-judge, which can be found in the appendix.
> > > >
> > > > Please consider raising our score if you find that the quality of our work has improved. We truly appreciate your time in reviewing our work.

---

### Meta-Review · Area_Chair_XYEv · 2026-01-13

**Summary:**

This paper introduces Verbal Technical Analysis (VTA), a hybrid framework that pairs an LLM-generated “verbal” reasoning trace with a latent time-series forecasting backbone, and trains the reasoning component with an RL-style objective (Time-GRPO) to align explanations with predictive performance. The work is well presented and empirically thorough, including multi-market evaluations, ablations, and a domain-expert study showing preference for the generated reasoning traces, plus a portfolio-level validation. The main weaknesses are that the algorithmic novelty is somewhat incremental, i.e., Time-GRPO as GRPO with an inverse-MSE reward. The reproducibility/fairness relies on strong baseline tuning and there is a lack of clearer evidence that reasoning causally improves forecasts. The rebuttal resolves these issues by adding perturbation experiments (adversarial noise/indicator removal) that degrade MSE, and clarifies contributions vs. related work such as TimeCAP. Overall, I recommend to accept this manuscript.

**Reviewer Concerns:**

The main concerns are that the algorithmic novelty is somewhat incremental, i.e., Time-GRPO as GRPO with an inverse-MSE reward. The reproducibility/fairness relies on strong baseline tuning and there is a lack of clearer evidence that reasoning causally improves forecasts.  The rebuttal process have resolved most of these concerns.

**Reviewer Scores:**

Two reviewers suggests they will increase their scores.

---

### Decision · Program_Chairs · 2026-01-26

Accept (Poster)